# Structural basis for VLDLR recognition by eastern equine encephalitis virus

Pan Yang [1,6], Wanyu Li [1,6], Xiaoyi Fan [1], Junhua Pan [1,2], Colin J. Mann[1], Haley Varnum [1], Lars E. Clark[1], Sarah A. Clark[1], Adrian Coscia[1], Himanish Basu [3], Katherine Nabel Smith [1], Vesna Brusic[1] & Jonathan Abraham [1,4,5] ✉

Eastern equine encephalitis virus (EEEV) is the most virulent alphavirus that infects humans, and many survivors develop neurological sequelae, including paralysis and intellectual disability. Alphavirus spike proteins comprise trimers of heterodimers of glycoproteins E2 and E1 that mediate binding to cellular receptors and fusion of virus and host cell membranes during entry. We recently identified very-low density lipoprotein receptor (VLDLR) and apolipoprotein E receptor 2 (ApoER2) as cellular receptors for EEEV and a distantly related alphavirus, Semliki Forest virus (SFV). Here, we use single-particle cryo-electron microscopy (cryo-EM) to determine structures of the EEEV and SFV spike glycoproteins bound to the VLDLR ligand-binding domain and found that EEEV and SFV interact with the same cellular receptor through divergent binding modes. Our studies suggest that the ability of LDLR-related proteins to interact with viral spike proteins through very small footprints with flexible binding modes results in a low evolutionary barrier to the acquisition of LDLR-related proteins as cellular receptors for diverse sets of viruses.

Alphaviruses are enveloped RNA viruses that cause diseases in humans that include acute febrile illness with rash and arthritis or lethal encephalitis. Encephalitic alphaviruses include EEEV, Venezuelan equine encephalitis virus (VEEV), and western equine encephalitis virus. These alphaviruses are transmitted by mosquitoes and cause outbreaks of encephalitis in horses and humans in the Americas[1]. The case fatality rate of EEEV infection can be as high as 75%[2], and there are no approved drugs or vaccines against EEEV. Additionally, up to 90% of individuals that survive symptomatic EEEV infection develop neurological sequalae[2]. EEEV outbreaks continue to occur at unpredictable intervals, and in years to come, EEEV may spread into new areas because of climate change and environmental perturbations[1].

In addition to serving as cellular receptors for EEEV, VLDLR and ApoER2 are also cellular receptors for SFV and Sindbis virus (SINV)[3],

two Old World alphaviruses that cause febrile illnesses in humans[4–6]. VLDLR and ApoER2 are canonical members of the LDLR-related receptor family, and both are expressed in the brain[7]. Their ligand-binding domains, which contains sites for alphavirus attachment[3], are found in the membrane-distal, N-terminal region of the receptors, and contain a series of LDLR class A (LA) repeats that are sequentially arranged like beads on a string (Fig. 1a). VEEV does not bind VLDLR or ApoER2 as cellular receptors and instead binds another LA repeat-containing membrane protein that is expressed in the brain: low-density lipoprotein receptor class A domain containing 3 (LDLRAD3)[3,8]. Importantly, while LDLRAD3 is a receptor for VEEV, it is not a receptor for any other tested alphavirus[3,8]. Furthermore, the VEEV E2–E1 residues that make critical contacts with LDLRAD3 are not conserved in other alphaviruses[8–10]. Thus, the previously available

[1]Department of Microbiology, Blavatnik Institute, Harvard Medical School, Boston, MA, USA. [2]Biomedical Research Institute and School of Life and Health Sciences, Hubei University of Technology, Wuhan, Hubei, China. [3]Department of Immunology, Blavatnik Institute, Harvard Medical School, Boston, MA, USA. [4]Department of Medicine, Division of Infectious Diseases, Brigham & Women's Hospital, Boston, MA, USA. [5]Center for Integrated Solutions in Infectious Diseases, Broad Institute of Harvard and MIT, Cambridge, MA, USA. [6]These authors contributed equally: Pan Yang, Wanyu Li. ✉e-mail: jonathan_abraham@hms.harvard.edu

cryo-EM structures of VEEV bound to LDLRAD3[9,10] do not explain how other alphaviruses bind to LA repeats.

## Results

### Alphavirus spike proteins recognize different LA repeats

The ligand-binding domain of VLDLR has eight LA repeats (Fig. 1a). To determine if a single LA repeat can support EEEV, SFV, and SINV E2−E1 mediated cellular entry, we generated constructs in which the eight VLDLR LA repeats were replaced by a single LA repeat (Fig. 1a). We overexpressed these truncated VLDLR variants or full length VLDLR in K562 cells, a human lymphoblast cell line that does not express VLDLR or ApoER2[3], and confirmed cell surface expression of constructs using immunostaining, although the single LA2 repeat construct had lower expression (Supplementary Fig. 1a, b). Reporter virus particles (RVPs) are a single-cycle system derived from a Ross River virus replicon that contain properly processed heterologous alphavirus spike proteins and can be used to study alphavirus entry[3]. We used EEEV, SFV, and SINV RVPs that express green fluorescent protein (GFP) to infect K562 cells transduced with single LA repeat constructs and found that each RVP differed in the repeats it could recognize to infect cells under tested conditions (Fig. 1b and Supplementary Fig. 1c−f). EEEV RVPs could infect K562 cells expressing the broadest range of LA repeats (LA1, LA2, LA3, LA5, or LA6) (Fig. 1b and Supplementary Fig. 1d). Only three out of the eight VLDLR LA repeats (LA4, LA7, and LA8) could not support infection by any of the RVPs we tested, suggesting that these particular LA repeats lack critical determinants required for interaction with alphavirus spike proteins (Fig. 1b and Supplementary Fig. 1d−f). Thus, despite their shared ability to bind VLDLR as a cellular receptor[3], EEEV, SFV, and SINV all differ in the LA repeats they can recognize.

### Cryo-EM structure of EEEV bound to VLDLR

To visualize how EEEV interacts with LA repeats, we next sought to determine the cryo-EM structure of EEEV virus-like particles (VLPs) bound to VLDLR. To identify a construct suitable for structural analysis, we first tested receptor Fc fusion proteins for binding to immobilized EEEV VLPs using biolayer interferometry (Supplementary Fig. 2). An Fc fusion protein that contains the ectodomain of matrix remodeling associated protein 8 (MXRA8), a cellular receptor for the Old World alphavirus Chikungunya virus (CHIKV)[11], which we used as a negative control in our assays, did not bind to immobilized EEEV VLPs (Supplementary Fig. 2c, e). An Fc fusion protein that only contains LA1, VLDLR$_{LA1}$-Fc, bound EEEV VLPs with very low affinity (apparent K$_D$ of 15.5 μM) (Supplementary Fig. 2e, f). However, an Fc fusion protein that contains the full ligand-binding domain, VLDLR$_{LBD}$-Fc, bound VLPs tightly (apparent K$_D$ of 2.1 nM), likely because its multiple compatible LA repeats provided avidity, as demonstrated by the slower off rate (Supplementary Fig. 2e). We thus chose VLDLR$_{LBD}$-Fc for structural analysis.

To determine the structure of VLDLR$_{LBD}$-Fc-bound to EEEV VLPs, we used the block-based approach[12], which involves subparticle extraction, and focused our analysis on the quasi-threefold (q3) spikes. Because of variable occupancy and because maps likely reflected a mixture of different LA repeats bound to clefts on VLPs, structural analysis required collection of large datasets with extensive 3D classification to identify a subset of spikes that contained strong density for at least one of the bound LA repeats (Supplementary Fig. 3). The global resolution of q3 spikes focused on this receptor-bound cleft was 3.5 Å. For model building and interpretation, based on our mapping studies of EEEV−VLDLR LA repeat interactions (Fig. 1b and Supplementary Fig. 1d), we used the VLDLR LA1 sequence. This is because LA1 is located at the N-terminus of the VLDLR ectodomain and, in principle, would be the most accessible for EEEV E2−E1 binding. However, as noted above, the cryo-EM density we observed likely reflected a mixture of different LA repeats.

The cryo-EM structure revealed that VLDLR LA repeats bind clefts formed by adjacent E2−E1 protomers on EEEV (Fig. 1c). LA repeats have one face that contains a highly conserved calcium ion (Ca$^{2+}$) binding site with a neighboring aromatic residue that usually makes critical contacts with ligands, and another face that is less conserved and contains a short β-hairpin[9,10,13–16]. The aromatic residue is a tryptophan in all of the VLDLR LA repeats that can facilitate EEEV E2−E1-mediated entry when expressed as single LA repeats (e.g., LA1 W50) (Fig. 2a). In the structure of the complex, two basic residues in EEEV E2, K156 and R157, make the most prominent contacts with VLDLR (Fig. 2b). Both are found in the central arch of the E2 β-ribbon connector[17], an E2 subregion that folds from two polypeptide segments that interrupt the connections between E2 domains A and B, and B and C (Fig. 1d and Supplementary Fig. 5a). The aliphatic portion of the EEEV E2 K156 side chain makes hydrophobic contacts with LA1 W50, and the E2 K156 side chain ε-amino group interacts with the side chains of LA1 D53 and D57 (Fig. 2b). E2 R157 interacts with the side chain of LA1 D55 and, nearby, the H155 side chain interacts with the LA1 D57 side chain (Fig. 2b). The adjacent spike protein protomer (E2'−E1') seems to make minimal contacts with the face of the LA repeat that contains the short β-hairpin and is less conserved, and these interactions only involve backbone atoms on the receptor (Supplementary Fig. 5b). However, our ability to visualize side chain contacts that may occur with individual LA repeats in this region is limited by the fact that this face of the LA repeat is not well conserved and the density we observed likely represents a mixture of different LA repeats.

To determine whether VLDLR binding causes conformational changes in EEEV E2−E1, we also determined the cryo-EM structure of unliganded EEEV VLPs and determined maps of unliganded q3 spikes (resolution of 3.0 Å) (Supplementary Figs. 4b and 6). During receptor binding, the loop that contains H155, K156, and R157 in the central arch of the E2 β-ribbon connector undergoes a small conformational change (Supplementary Fig. 5c). There are otherwise no conformational changes elsewhere on the EEEV spike glycoprotein.

### Functional assessment of EEEV−VLDLR interactions

To study the importance of VLDLR LA repeat contacts with EEEV E2−E1, we next conducted infectivity assays with EEEV RVPs and K562 cells that overexpress wild type or mutant LA1 single LA repeat constructs. We confirmed expression of all constructs using cell surface immunostaining (Supplementary Fig. 7a). LA1 residue W50 either remains a tryptophan or is replaced by a phenylalanine, a lysine, or an arginine in the other VLDLR LA repeats (Fig. 2a). An LA1 W50F substitution did not have an effect on EEEV RVP entry (Fig. 2c), suggesting that a phenylalanine could still establish hydrophobic interactions with the aliphatic portion of the E2 K156 side chain required for LA repeat engagement. However, substituting LA1 W50 with an arginine abolished EEEV RVP entry (Fig. 2c). Additionally, single residue mutations in the cluster of LA repeat acidic residues that is involved in Ca$^{2+}$ coordination (LA1 mutations D53A, D55I, and D57A) all abolished EEEV RVP entry into cells (Fig. 2c).

To test the importance of individual EEEV E2−E1 residues to interactions with VLDLR, we conducted cell surface staining experiments with VLDLR$_{LBD}$-Fc and HEK293T cells transfected with plasmids encoding wild-type or mutant spike proteins (EEEV E3−E2−[6K/TF]−E1). We observed comparable cell surface expression on transfected HEK293T cells with immunostaining using an anti-EEEV E1 monoclonal antibody (Supplementary Fig. 7c). VLDLR$_{LBD}$-Fc bound cells expressing wild-type EEEV spike proteins, but not cells expressing mutant spike proteins that contain the K156A or R157A substitutions in E2 (Fig. 2d and Supplementary Fig. 7c, d). VLDLR$_{LBD}$-Fc did not bind HEK293T-cells expressing the CHIKV spike proteins, used as a negative control in the binding assays, although these cells could be recognized by the MXRA8$_{ect}$-Fc fusion protein (Fig. 2d). Therefore, EEEV E2 residues K156

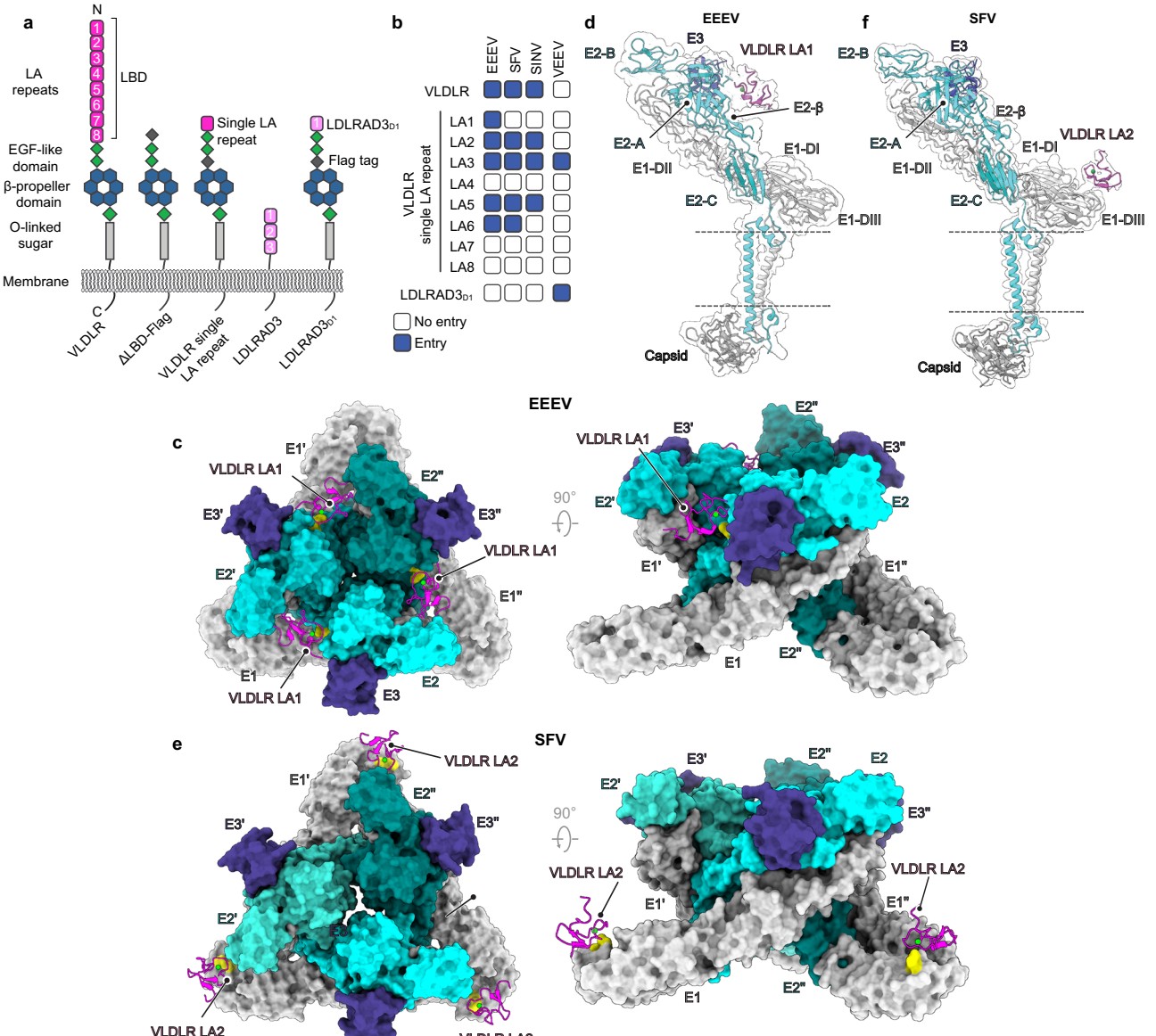

**Fig. 1 | EEEV and SFV use distinct binding modes to recognize VLDLR.**
**a** Constructs used to map alphavirus receptor LDLR class A (LA) repeat binding preferences. LDLRAD3 is a VEEV receptor[8]. VLDLR N- and C-termini are indicated. LBD ligand-binding domain. **b** Summary of results obtained when GFP-expressing reporter virus particles (RVPs) for the indicated alphaviruses were used to infect K562 cells expressing the receptor constructs shown in (**a**), with entry quantified using flow cytometry. See Supplementary Fig. 1 for additional information. "Entry" indicates that the specified construct mediates statistically significantly higher ($p < 0.05$) RVP infection than the control ΔLBD construct. **c** Top (left panel) and side (right panel) views of the cryo-EM structure of the EEEV spike protein bound to VLDLR$_{LBD}$-Fc. The spike protein is shown in surface representation and LA repeats

are shown as ribbon diagrams. E2 residues K156 and R157 are shown in yellow. $Ca^{2+}$ ions are shown as green spheres. **d** Ribbon diagram and cryo-EM density of the EEEV spike protein bound to VLDLR$_{LBD}$-Fc with an isolated view of a single protomer. **e** Top (left panel) and side (right panel) views of the cryo-EM structure of the SFV spike protein bound to VLDLR$_{LBD}$-Fc. E1 residues K345 and K347 are shown in yellow. **f** Ribbon diagram and cryo-EM density of the SFV spike protein bound to VLDLR$_{LBD}$-Fc with an isolated view of a single protomer. In (**d**, **f**), E2 domains (A, B, and C) and E1 domains (DI–III) and the β-ribbon connector (E2-β) are indicated. Dashed lines indicate the position of the viral membrane. The associated capsid protomer is also shown.

and R157 are critical determinants of the interaction with VLDLR LA repeats.

### Comparison with the SFV–VLDLR binding mode

SFV is another alphavirus that binds VLDLR as a cellular receptor[3]. However, EEEV E2 residues K156 and R157, which EEEV uses to make critical contacts with LA repeats (Fig. 2b, d), are not conserved in SFV E2 and are instead replaced by threonines (Supplementary Fig. 8a). This lack of conservation suggests that EEEV and SFV use different binding surfaces to interact with VLDLR. To visualize how SFV engages VLDLR, we determined the cryo-EM structure of SFV VLPs bound to

VLDLR$_{LBD}$-Fc (resolution of 4.9 Å) (Supplementary Fig. 9). Maps of VLDLR$_{LBD}$-Fc-bound SFV VLPs did not reveal density for LA repeats in the cleft between adjacent SFV E2–E1 protomers, but they instead revealed additional density near the base of E1 at the two-fold and five-fold icosahedral symmetry axes (Supplementary Fig. 9). We used the block-based approach[12] and focused our analysis on five-fold blocks. We obtained a 3.8 Å map focused on the LA repeat binding site, which allowed us to dock and build an LA repeat that interacts with domain III of E1 (Fig. 1e, f and Supplementary Fig. 9). For the purpose of model building and interpretation, we used the VLDLR LA2 sequence, because our functional assays suggested that VLDLR LA1 is not

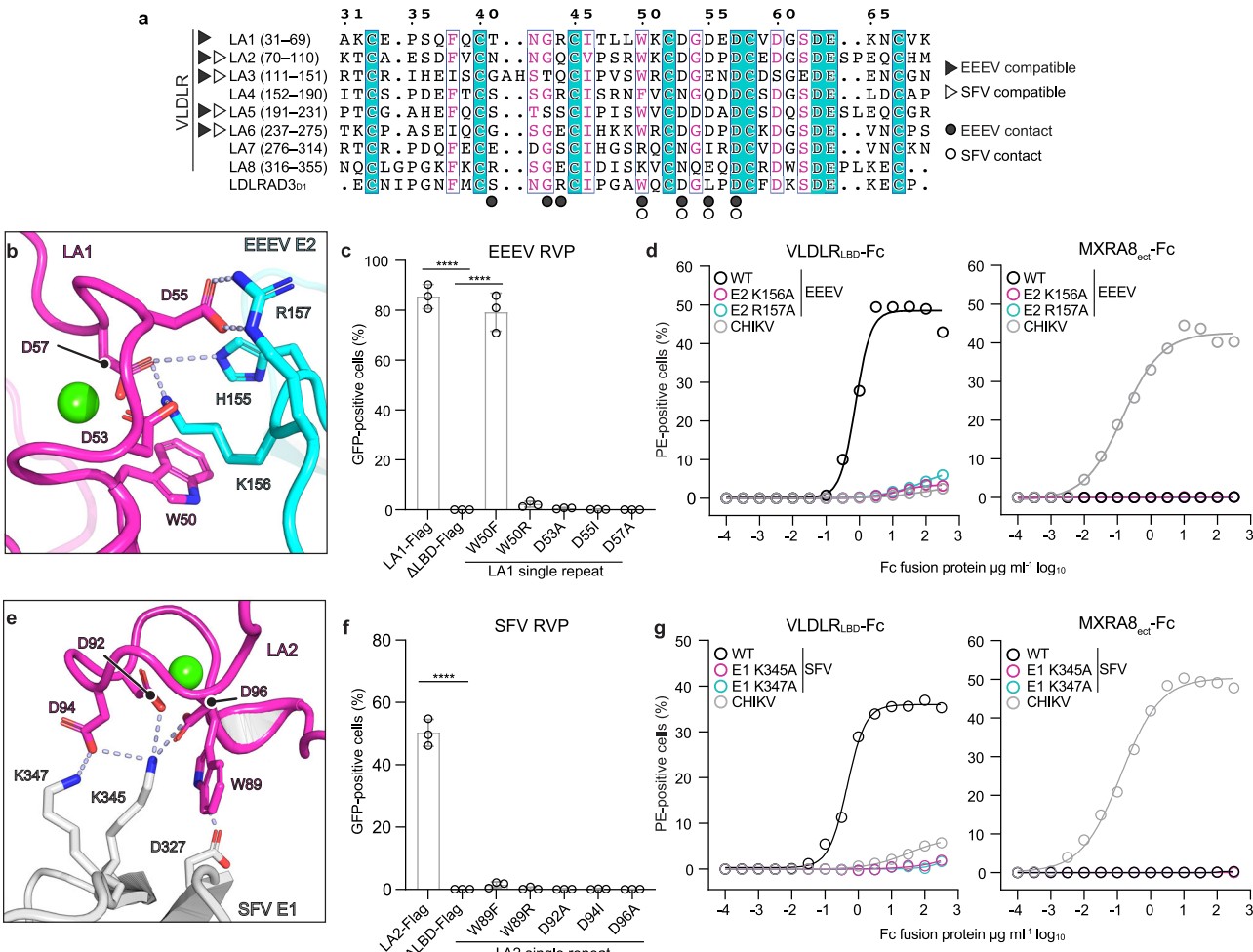

**Fig. 2 | EEEV E2 and SFV E1 spike glycoprotein interactions with VLDLR.**
**a** Alignment of VLDLR LA repeat and Domain 1 (D1) of the VEEV receptor LDLRAD3[8] amino acid sequences, generated using ESPript3[59]. Amino acid numbering, shown in parentheses, is based on full-length protein sequences (VLDLR GenBank NP_003374.3, LDLRAD3[8] GenBank AAH42754.2). Cyan background highlights residues that are completely conserved in all LA repeats. Boxed residues highlight positions where a single majority residue or multiple chemically similar residues could be identified. Such residues are highlighted in magenta. The panel was generated using ESPript3[59]. **b** Zoom in view of the interface between EEEV E2 and VLDLR LA1. **c** EEEV FL91-469 RVP infection of K562 cells expressing wild-type or mutant single VLDLR LA repeat constructs with entry quantified using flow cytometry for GFP. **d** VLDLR$_{LBD}$–Fc or MXRA8$_{ect}$–Fc immunostaining of HEK293T cells transfected with constructs encoding wild-type or mutant EEEV FL91-469 or wild-type CHIKV E3–E2–[6K/TF]–E1 (See Supplementary Fig. 7c). **e** Zoom in view of the interface between SFV E1 and VLDLR LA2. **f** SFV RVP infection of K562 cells expressing wild-type and mutant single VLDLR LA2 constructs with entry quantified using flow cytometry for GFP. **g** VLDLR$_{LBD}$–Fc or MXRA8$_{ect}$–Fc immunostaining of HEK293T cells transfected with constructs encoding wild-type or mutant SFV or CHIKV E3–E2–[6K/TF]–E1 (see Supplementary Fig. 7e). Data are mean ± s.d. from three experiments performed in duplicate or triplicate ($n = 3$ independent experiments) in (**c**, **f**), and two experiments performed in triplicate ($n = 2$ independent experiments) in (**d**, **g**). One-way ANOVA with Dunnett's multiple comparisons test, ****$p < 0.0001$ compared to ΔLBD-FLAG (**c**, **f**). PE: R-phycoerythrin. Ca$^{2+}$ ions are shown as green spheres in (**b**, **e**). Source data are provided as a Source Data file.

compatible with SFV E2–E1 binding, and LA2 is otherwise located closest to the N-terminus of the VLDLR ectodomain and may be the most accessible for SFV E2–E1 binding (Fig. 1b and Supplementary Fig. 1e). However, the density likely reflected an average of different compatible LA repeats wrapped around the five-fold axis.

LA repeat interactions with SFV center on E1 residues K345 and K347 (Fig. 2e). The aliphatic portion of the SFV E1 K345 side chain contacts the side chain of LA2 W89, and the ε-amino group of the K345 side chain interacts with LA2 D92, D94, and D96 (Fig. 2e). Additionally, the ε-amino group of E1 K347 interacts with LA2 D94, an LA repeat residue whose backbone carbonyl coordinates the calcium (Fig. 2e). Nearby, the side chain of SFV E1 D327 interacts with the indole nitrogen of LA2 W89 (Fig. 2e).

To study the importance of the visualized VLDLR LA repeat contacts with SFV E1, we used infectivity assays with SFV RVPs and K562 cells that overexpress wild-type or mutant LA2 single repeat constructs. The LA2 W89F substitution abolished entry of SFV RVPs, as did

a W89R substitution (Fig. 2f and Supplementary Fig. 7b). Additionally, single residue mutations in the cluster of LA repeat acidic residues that is involved in Ca$^{2+}$ coordination (LA2 mutations D92A, D94I, and D96A) all abolished SFV RVP entry into cells (Fig. 2f and Supplementary Fig. 7b). To test the importance of SFV E1 residues K345 and K347 to interactions with VLDLR, we transfected HEK293T cells with plasmids encoding wild-type SFV E3–E2–(6K/TF)–E1 or constructs in which these E1 residues were substituted to alanine and confirmed expression using immunostaining (Supplementary Fig. 7e). VLDLR$_{LBD}$–Fc could not stain cells expressing the E1 K345A or K347A mutant constructs (Fig. 2g), indicating that E1 residues K345 and K347 are critical to the SFV–VLDLR interaction. Interestingly, these two basic residues are not conserved in EEEV E1, and instead are replaced by a threonine and an alanine (T346 and A348) (Supplementary Fig. 8b).

Our comparative structural and functional analysis of EEEV and SFV receptor-binding modes therefore reveals that both of these alphaviruses evolved drastically different binding surfaces for the

same cellular receptor (Fig. 1c–f). EEEV interactions with VLDLR focus on two basic residues on the E2 subunit of the spike glycoprotein, while SFV interactions center on two basic residues on the E1 subunit of the spike glycoprotein. The relaxed requirement of two basic residues on the viral glycoproteins, positioned such that they can effectively engage the aromatic and acidic residues in the LA repeat $Ca^{2+}$ coordination center, while surrounded by a small, compatible interaction footprint explains the ability of the EEEV and SFV E2–E1 glycoproteins to recognize a broad array of LA repeats (Fig. 1b and Supplementary Fig. 1d, e). As a consequence of a binding mode that involves only a small number of residues on each of the interaction partners, mutating single residues on the viral spike proteins or receptor in almost every case we tested abrogated interactions (Fig. 2c, d, f, g).

Interestingly, EEEV RVPs could enter cells that express LA1 with the W50F mutation (Fig. 2c), suggesting that EEEV can engage LA repeats that contain either a tryptophan or a phenylalanine as the key aromatic residue. However, the LA2 W89F mutation abolished SFV RVP entry into K562 cells (Fig. 2f). These findings suggest that the polar contact the SFV E1 D327 side chain makes with the indole nitrogen of the LA repeat tryptophan (Fig. 2e) is a critical interaction.

### E3 hinders multiple LA repeat binding modes

While our structural studies allowed us to visualize a single LA repeat bound to the EEEV spike protein, recent studies reported that the EEEV spike protein have up to three binding sites for VLDLR LA repeats[18,19]. Site 1 is located in clefts formed between adjacent spike protein protomers as identified in our study, site 2 is located on top of E2 domain A close to the E3 binding site, and site 3 is located in E2 domain B[18,19]. Interestingly, site 3 is only present in two strains of EEEV, including in EEEV strain PE6 that we used to generate VLPs that are also used in the development of a vaccine under clinical investigation (Supplementary Fig. 8a)[18,19]. LA repeat contacts at sites 1–3 are similar: calcium-coordinating aspartic residues as well as a nearby tryptophan on the LA repeats contact independent clusters of two basic residues on the EEEV E2 spike protein[18,19].

Importantly, E3 density is present in our maps, unlike in the structures reported by Adams et al.[18] and Cao et al.[19] or in prior structural studies of EEEV (Supplementary Fig. 3)[20–22]. Overlaying the EEEV VLP-VLDLR LBD atomic coordinates (PDB: 8UFB)[18] from Adams et al., which contains VLDLR LA repeats at sites 1–3, revealed that the presence of E3 would likely sterically hinder LA repeat binding at site 2 (Fig. 3a, b). Partial obscuring of the site 2 LA repeat binding site may prevent the VLDLR ligand-binding domain from wrapping around E2 to allow a third LA repeat to engage site 3. Thus, the presence of E3 in our VLP samples likely prevented us from being able to visualize a three LA repeat-binding mode on the EEEV spike protein. As previous structural studies found that retainment of E3 in infectious CHIKV particles affects binding of the CHIKV receptor MXRA8[23], our findings add to a potential role of virion bound E3 in modulating alphavirus receptor binding.

### Spike protein determinants of VLDLR and ApoER2 binding

Aside from VLDLR, ApoER2 is also a functional receptor for EEEV[3]. While there are multiple splice variants of ApoER2, shorter isoforms of ApoER2 are the dominant forms expressed in the brain[24–26]. We previously showed that a human ApoER2 isoform that contains only three LA repeats in its ligand-binding domain (ApoER2 iso2) (Fig. 3c) can serve as an EEEV receptor[3]. The spike protein of EEEV strain FL91-469, like most EEEV strains, only contains LA repeat binding sites 1 and 2 (Supplementary Fig. 8a). We generated mutant RVPs in which the EEEV FL91-469 spike protein was mutated to contain alanine substitutions in key basic residues of LA repeat binding sites 1 (K156A) or 2 (K231A/K232A) (Supplementary Fig. 7f). Wild-type EEEV FL91-469 RVPs robustly infected K562 cells expressing human VLDLR

or ApoER2 iso2 (Fig. 3d). The FL91-469 E2 site 1 (K156A) mutant RVPs were unable to bind either VLDLR or ApoER2 iso2 to infect K562 cells. The FL91-469 E2 site 2 (K231A/K232A) mutant RVPs showed diminished entry on K562 cells expressing VLDLR but showed no defect in their ability to infect K562 cells expressing ApoER2 iso2 (Fig. 3d). These findings suggest that for EEEV strains that can only bind LA repeats at sites 1 and 2, both VLDLR and ApoER2 binding modes critically depend on contacts K156 makes with LA repeats on site 1.

EEEV strain PE6 has three LA repeat binding sites (Fig. 3b and Supplementary Fig. 8a). EEEV PE6 RVPs robustly infected K562 cells expressing VLDLR or ApoER2 iso2 (Fig. 3e and Supplementary Fig. 7f). However, unlike with experiments with EEEV FL91-469 RVPs, site 1 mutant (K156A) PE6 RVPs could still robustly infect K562 cells expressing VLDLR, consistent with previous findings[18]. Mutation of site 2 had no effect on the ability of EEEV PE6 RVPs to infect K562 cells expressing VLDLR and ApoER2 iso2 but did decrease mutant RVP entry into K562 cells expressing ApoER2 iso2 (Fig. 3d and Supplementary Fig. 7f). Together, our findings suggest that for the more rare EEEV strains that have spike proteins that can bind LA repeats at three sites, VLDLR and ApoER2 binding can be achieved by engagement of any two available sites; ApoER2 iso2 binding, however, nonetheless still heavily relies on site 1.

We also tested the requirement of the SFV residue that contacts VLDLR, E1 K345 (Fig. 2e), in SFV RVP entry. As expected, SFV RVPs bearing the E1 K345A mutation could not infect K562 cells overexpressing VLDLR or ApoER2 iso2 (Supplementary Fig. 7g), suggesting that SFV critically depends on E1 K345 to recognize the calcium-coordinating aspartate residues in the LA repeats of both VLDLR and ApoER2 iso2.

### LA repeat binding sites are determinants of neuronal entry

EEEV tropism during brain infection is directed towards neurons, and EEEV causes encephalitis in mice[27–29]. We next tested whether EEEV E2 basic residues in site 1 (E2 K156) or site 2 (E2 K231 and K232) are required for infection of primary murine cortical neurons isolated from embryonic mice as a physiologically relevant cell type. While wild-type EEEV FL91-469 RVPs robustly infected murine neurons, the E2 K156A substitution alone dramatically decreased neuronal entry (Fig. 3f, g), indicating that LA repeat binding site 1 is crucial for EEEV entry into primary neurons. LA repeat binding site 2 mutations (K231A and K232A) likewise abolished RVP entry into murine neurons (Fig. 3f, g). These data indicate that the EEEV E2 LA-repeat binding sites 1 and 2 are critical determinants of EEEV neurotropism.

### Calcium is necessary for EEEV and SFV VLDLR binding

Our structural and mutational analyses suggests that calcium coordinating amino acids in VLDLR LA repeats are essential for interactions with the EEEV and SFV spike proteins. To test whether calcium chelation would impact VLDLR binding to EEEV and SFV, we used biolayer interferometry experiments with receptor Fc fusion proteins and VLPs performed in the presence or absence of ethylenediaminetetraacetic acid (EDTA). Addition of EDTA during the association phase of binding prevented $VLDLR_{LBD}$-Fc binding to immobilized EEEV or SFV VLPs (Supplementary Fig. 10a). Similarly, addition of the ligand antagonist receptor-associated protein (RAP), used as a control, to $VLDLR_{LBD}$-Fc during the association phase of binding prevented the Fc fusion protein from binding to immobilized EEEV and SFV VLPs (Supplementary Fig. 10a). Conversely, EDTA treatment did not impact the ability of a CHIKV receptor Fc fusion protein ($MXRA8_{ect}$-Fc) to bind CHIKV VLPs (Supplementary Fig. 10b), which was expected as the CHIKV-MXRA8 interaction is not suspected to be calcium-dependent[23,30]. These results indicate that calcium ions are required for VLDLR binding to the EEEV and SFV spike proteins.

## Comparison of LA repeat interactions with other ligands

The general features of the interactions we observed between VLDLR LA repeats and alphavirus spike proteins are similar to those that have

been observed in how LDLR-related proteins interact with physiological ligands or other viral entry proteins (Fig. 4a and Supplementary Fig. 11a). These ligands include RAP, a chaperone for LDLR-related

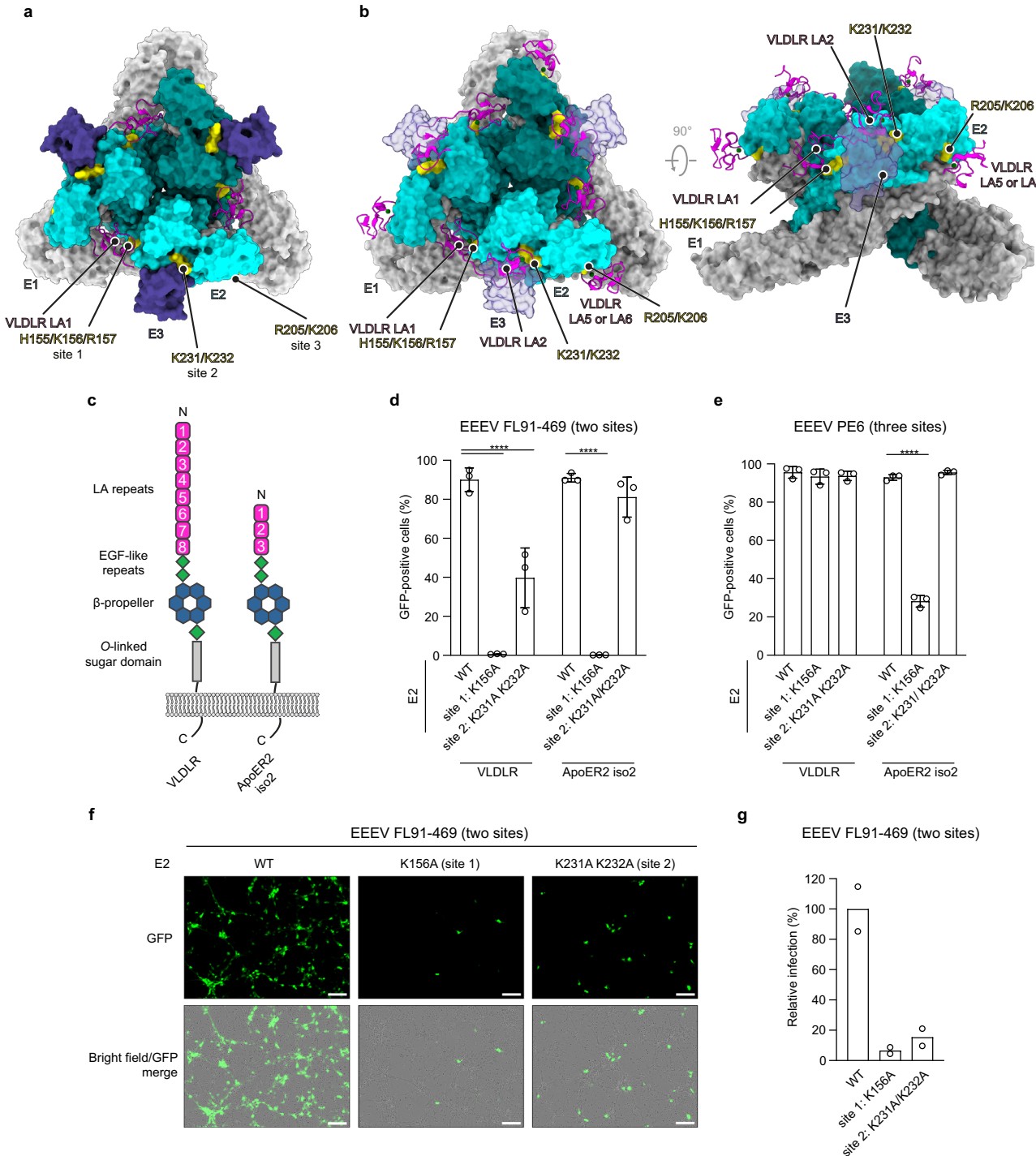

**Fig. 3 | The EEEV spike protein engages multiple LA repeats. a** Top view of the cryo-EM structure of the EEEV spike protein in complex with VLDLR LA1. E2 residues that contact VLDLR as determined by Adams et al.[18] are labeled and highlighted in yellow. **b** Top (left panel) and side (right panel) views of the cryo-EM structure of the EEEV spike protein (unliganded) determined in this study, fitted with VLDLR LA repeats bound to the EEEV spike as determined by Adams et al. (PDB: 8UFB)[18]. In (**a**, **b**), the spike protein is shown in surface representation and LA repeats are shown as ribbon diagrams. Ca²⁺ ions are shown as green spheres. **c** Schematic depiction of VLDLR and ApoER2 isoform 2 (ApoER2 iso2). LA repeats are numbered from the N terminus. **d**, **e** K562 cells expressing human VLDLR and ApoER2 iso2 were infected

with wild-type or mutant GFP-expressing RVPs for EEEV FL91-469 or EEEV PE6. Infection was monitored by flow cytometry. **f** Embryonic day 17 murine cortical neurons were infected with GFP-expressing wild-type or mutant EEEV FL91-469 RVPs. Infection was quantified using a live cell imaging system (see "Methods" for additional information). Representative GFP images and merged images of GFP and bright field are shown. Scale bars are 100 μm. **g** Quantification of neuronal infection in (**f**). Data are mean ± s.d. from three experiments performed in triplicates (*n* = 3 independent experiments) (**d**, **e**) or two experiments performed in triplicates (*n* = 2 independent experiments) (**g**). Two-way ANOVA with Dunnett's multiple comparisons test, ****$p < 0.0001$ (**d**, **e**). Source data are provided as a Source Data file.

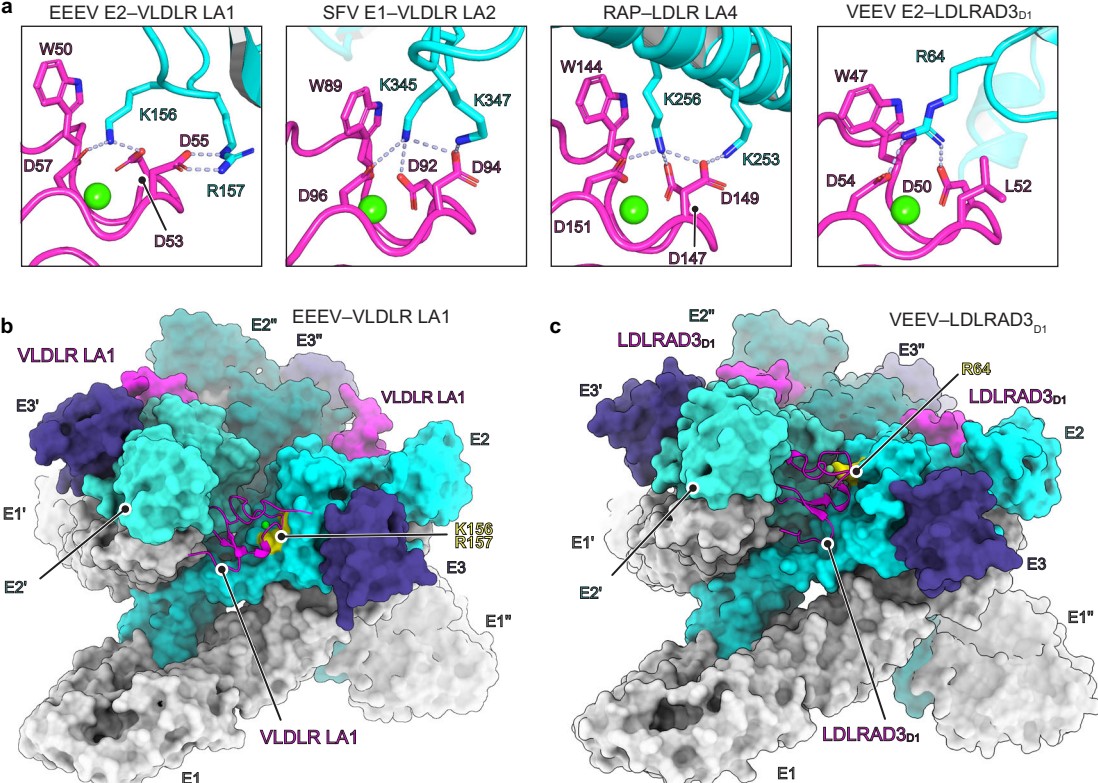

**Fig. 4 | Comparison of LA repeat interactions with physiological ligands and other viral entry proteins. a** Zoom in view of contacts basic residues EEEV E2, SFV E1, receptor-associated protein (RAP) (PDB ID: 2FCW)[35] and VEEV E2 (PDB ID: 7FFN)[9] make with the Ca²⁺-coordinating acidic residues and central aromatic residue in LA repeats. Viral glycoproteins and ligands are colored in cyan and LA repeats are colored in magenta. **b** Side view of the cryo-EM structure of EEEV bound to VLDLR_LBD-Fc. The complex is shown in surface representation with the exception of one of the VLDR LA repeats. Basic residues in EEEV E2 (K156 and R157) that anchor LA repeat interactions are shown in yellow. **c** Side view of the previously determined cryo-EM structure of VEEV bound to LDLRAD3_D1-Fc (PDB: 7FFN)[9], with one of the LA repeats shown as a ribbon diagram. A basic residue in VEEV E2 (R64) that anchors interactions with LA repeats is shown in yellow. Ca²⁺ ions are shown as green spheres.

proteins that binds LA repeats to block ligand attachment[31] and also blocks EEEV and SFV E2–E1 binding to VLDLR and ApoER2[3], and Reelin, a large secreted glycoprotein that helps orchestrate the development of laminated structures in the brain and binds the VLDLR and ApoER2 ligand-binding domains (Fig. 4a and Supplementary Fig. 11a)[7,14,32,33]. Other viral entry proteins that have similar general binding modes include that of vesicular stomatitis virus (VSV) glycoprotein G[15] and human rhinovirus 2 (HRV2) VP1 (Supplementary Fig. 11a)[34]. All of these interactions are anchored by an aromatic residue on the LA repeat that forms key hydrophobic contacts with the side chain of a lysine on the partner protein. The side chain of that lysine is then also encircled by Ca²⁺ coordinating acidic residues on the LA repeat (Fig. 4a and Supplementary Fig. 11a)[14,35]. In EEEV E2 and SFV E1 interactions with LA repeats, a second basic residue interacts with the sidechain of an acidic residue whose backbone carbonyl, but not side chain, is involved in Ca²⁺ coordination (D55 in LA1 and D94 in LA2) (Fig. 4a). When interactions are compared, the contacts that EEEV and SFV make with LA repeats are most similar to how RAP interacts with LA repeats, with RAP using clusters of two basic residues to interact with the LA repeat Ca²⁺ coordination center (Fig. 4a and Supplementary Fig. 11a).

## Comparison with VEEV–LDLRAD3 interactions

Although both VEEV and EEEV bind to LA repeats in clefts formed by adjacent E2–E1 protomers on assembled spike proteins, the basic residues on the VEEV E2 proteins that make critical receptor contacts are in different E2 subregions. VEEV E2 R64, which makes critical contacts with the aromatic residue and the Ca²⁺ coordinating acidic residues in LDLRAD3_D1[9] (Fig. 4a), is in E2 domain A, rather than in the

E2 β-ribbon connector, and is replaced by an alanine (A63) in EEEV E2 (Supplementary Fig. 5a). The altered positioning of the critical basic residue on VEEV vs. EEEV E2 allows LDLRAD3_D1 to fit much more snugly within the cleft, with the adjacent VEEV E2′–E1′ protomer making more contacts with the opposite face of LDLRAD3_D1 (Fig. 4b, c). As a result, LDLRAD3_D1 has an interaction surface on VEEV E2–E1 that is almost twice as large as that of LA1 on EEEV E2–E1 (1111 Å² vs. 645 Å²) (Supplementary Fig. 11b). Such an extensive contact area may explain why VEEV binds LDLRAD3_D1 with high affinity (reported measurements are 58 nM and 351 nM)[9,10], in comparison to the weak binding we observed for VLDLR_LA1-Fc binding to EEEV VLPs (15.5 μM) (Supplementary Fig. 2e, f).

In our studies mapping VLDLR LA repeat interactions with alphavirus spike proteins, we also found that VEEV RVPs could infect K562 cells expressing a chimeric VLDLR receptor construct in which LDLRAD3_D1 replaces the VLDLR ligand-binding domain (Fig. 1b and Supplementary Fig. 1g). Unexpectedly, VEEV could also infect K562 cells expressing VLDLR LA3 as a single repeat (Fig. 1b and Supplementary Fig. 1g). Because VLDLR is not a receptor for VEEV[3], these findings suggest that steric hindrance from neighboring LA repeats might constrain VEEV's ability to bind LA repeat-containing receptors other than LDLRAD3[9]. VEEV thus seems highly adapted to binding to LDLRAD3_D1 and has the most restricted ability to recognize LA repeats; EEEV, in contrast, has evolved a more flexible binding mode that allows it to accommodate different LA repeats.

Despite their ability to bind multiple LA repeats found in VLDLR, EEEV and SFV do not bind LDLRAD3_D1 (Fig. 1b and Supplementary Fig. 1d, e)[3,8]. The cryo-EM structures provide an explanation for this

observation. In LDLRAD3$_{D1}$, a leucine (L52) replaces acidic residues (e.g., LA1 D55 and LA2 D94) whose side chains make salt bridges with basic residues on EEEV E2 and SFV E1 (Figs. 2a and 4a). The presence of a leucine at this position in LDLRAD3$_{D1}$ would remove these key salt bridges and compromise LA repeat interactions with the EEEV and SFV spike proteins. Indeed, substituting LA1 D55 or LA2 D94 with an iso-leucine (D55I or D94I), which is chemically similar to a leucine and would also remove the salt bridge, abolishes EEEV and SFV RVP entry into K562 cells expressing mutant single LA repeat constructs (Fig. 2c, f). These observations suggest that L52 in LDLRAD3 explains why LDLRAD3 is not a receptor for EEEV and SFV.

## Discussion

By determining cryo-EM structures of EEEV and SFV spike proteins bound to VLDLR LA repeats, we found that the spike proteins of these alphaviruses make similar contacts with LA repeats but use surfaces found on completely different subunits of their spike proteins (Fig. 1c–f). Taken together, these observations suggest that the two alphaviruses independently evolved the ability to bind LDLR-related proteins and may indicate a strong selective pressure for alphaviruses to maintain LDLR family members as entry receptors.

The LA repeat residues whose side chains contact the EEEV or SFV spike proteins are generally conserved in the LA repeats that can interact with EEEV or SFV E2–E1 (Figs. 1b and 2a and Supplementary Fig. 1d, e). These residues are also conserved in the N-terminal LA repeats of VLDLR orthologs we previously showed are functional receptors for EEEV or SFV in various capacities, including murine, equine, avian, mosquito (*Aedes albopictus* and *A. aegypti* lipophorin receptor 1) and worm (*Caenorhabditis elegans*) orthologs, and in ApoER2 orthologs (Supplementary Fig. 11c, d)[3]. Despite having N-terminal LA repeats that contain conserved key contact residues, not all the VLDLR orthologs we previously tested could serve as functional receptors for EEEV[3], suggesting that sequence variation in the less conserved regions of LA repeats also influence LA repeat dependencies. Indeed, our functional analysis revealed that the receptor-glycoprotein interface is exquisitely sensitive to perturbation (Fig. 2c, f). As such, even small changes in sequence elsewhere in the LA repeat could disrupt binding by introducing steric clashes or creating electrostatic repulsion with the rest of the spike protein.

The location of the SFV binding site for VLDLR, which involves domain III of E1 rather than in clefts between adjacent E2–E1 protomers, is unexpected, but is further supported by independently determined cryo-EM structures of SFV VLPs bound to VLDLR that were recently reported by Cao et al.[36]. For some of their high resolution structural analysis with SFV VLPs, Cao et al. used an Fc fusion protein that only contains VLDLR LA3, which improved the quality of maps by circumventing the issue of receptor density representing a mixture of different LA repeats bound to different clefts[36]. This allowed the authors to observe additional contacts made by the LA repeat that our maps did not reveal. Comparison of the VLDLR-bound SFV structures reveals that they there are in overall agreement, with slight rotation of the LA repeat on the binding site, which can be attributed to the different constructs used for structural analysis (e.g., VLDLR$_{LA3}$-Fc vs. VLDLR$_{LBD}$-Fc) (Supplementary Fig. 11e)[36].

Importantly, LDLR-related proteins are increasingly being recognized as cellular receptors for diverse sets of viruses in recent years; in addition to their roles as receptors for VSV[15,37] and minor group rhinoviruses (e.g., HRV2)[38], LDLR-related proteins have been described as receptors for the phlebovirus Rift Valley fever virus[39], the orthobunyavirus Oropouche virus[40], and attenuated strains of the morbillivirus canine distemper virus[41]. The observation that EEEV and SFV, two alphaviruses with structurally homologous spike gly-coproteins, have evolved distinct mechanisms to bind the same cellular receptor suggests that acquiring the ability to bind LDLR-related receptors may have a relatively low evolutionary barrier for viruses. The key determinant of binding on the viral spike protein is an exposed basic residue (lysine or arginine) that can interact with a cluster of relatively conserved acidic residues and an aromatic residue on the receptor, in addition to peripheral contacts that contribute to stabilizing the interaction. Avidity afforded by multiple compatible LA repeats within receptor ligand-binding domains and multiple copies of the receptors on cells then promotes tight binding of the viruses. This binding strategy is remarkably similar to that of physiological ligands for LDLR-related proteins, which also involve metal-dependent electrostatic recognition of exposed basic residues together with avidity effects resulting from the use of multiple sites[35].

In addition to serving as receptors for EEEV and SFV, VLDLR can also serve as a receptor for SINV[3]. The structures suggest that SINV will use a VLDLR binding mode that is distinct from either EEEV or SFV. EEEV E2 K156 and R157, the site 1 basic residues that are critical to the interaction of EEEV with VLDLR LA repeats, are respectively replaced by a leucine and a lysine in SINV E2 (Supplementary Fig. 8a). SFV E1 domain III residues K345 and K347, which are critical to the interaction of SFV with LA repeats, are respectively replaced by a histidine and a leucine in SINV E2 (Supplementary Fig. 8b). Because our functional analysis suggests that at least two spike protein basic residues are required to productively engage VLDLR LA repeats, and neither site in the SINV spike protein would meet this requirement, the SINV spike protein VLDLR binding mode is likely distinct than that of the EEEV and SFV spike proteins.

Interestingly, alphaviruses belonging to the western equine encephalitis complex, including SINV, were recently shown to recognize avian but not mammalian orthologs of MXRA8, the well-established CHIKV receptor[11], with a binding mode that is remarkably distinct from how CHIKV binds mammalian MXRA8[42]. These findings, along with our study, add to a growing understanding of the substantial structural plasticity through which alphavirus spike proteins can interact with cellular receptors on host cells[18,19,36,42].

Flexible receptor-binding modes that capitalize on very small footprints on viral spike proteins and avidity effects could also facilitate virus transmission across evolutionary divergent organisms, such as humans and arthropod vectors in the case of alphaviruses[43]. The multiple favorable structural and functional properties of LDLR-related proteins could in part explain why these molecules are recurrently targeted by closely and more distantly related viruses to initiate cellular entry.

## Methods

### Cells and viruses

We maintained HEK293T (human kidney epithelial, ATCC CRL-11268), 293FT (Thermo Fisher Scientific), and Vero E6 cells (*Cercopithecus aethiops* kidney epithelial, ATCC CRL-1586) in Dulbecco's modified Eagle's medium (DMEM, Gibco) supplemented with 10% (v/v) fetal bovine serum (FBS), 25 mM HEPES (Thermo Fisher Scientific), and 1% (v/v) penicillin-streptomycin (Thermo Fisher Scientific). We maintained Expi293F™ cells in Expi293™ Expression Medium (Thermo Fisher Scientific). We maintained K562 cells (human chronic myelogenous leukemia, ATCC CCL-243) in RPMI 1640 medium (Thermo Fisher Scientific) supplemented with 10% (v/v) FBS and 1% (v/v) penicillin-streptomycin (Thermo Fisher Scientific). We maintained C6/36 (*Aedes albopictus* larval tissue, provided by N. Perrimon) in Schneider's *Drosophila* Medium (Thermo Fisher Scientific) supplemented with 10% (v/v) FBS, 1% (v/v) non-essential amino acids (NEAA, Thermo Fisher Scientific), and 1% (v/v) penicillin-streptomycin. We confirmed the absence of mycoplasma in all cell lines through monthly testing using an e-Myco PCR detection kit (Bulldog Bio #25234). Cell lines were not authenticated.

## Reporter virus particle generation

Reporter virus particles (RVPs) were generated by transfecting HEK293T cells with a modified pRR64 plasmid[44] and a pCAGGS vector expressing the heterologous alphavirus E3–E2–(6K/TF)–E1 proteins with a start codon upstream of E3[3]. The alphavirus pCAGGS E3–E2–(6K/TF)–E1 plasmids we used to generate RVPs encode the proteins of EEEV (Florida 91-469, GenBank: Q4QXJ7.1; PE6, GenBank: AY722102.1), SFV (SFV4, GenBank: AKC01668.1), SINV (Toto1101 T6P144, GenBank: AKZ17594.1), and VEEV (TC-83, GenBank: AAB02517.1). We used Lipofectamine 3000 (Invitrogen) to transfect 293FT cells using the manufacturer's protocol and replaced media with Opti-MEM™ (Thermo Fisher Scientific) supplemented with 5% (v/v) FBS, 25 mM HEPES, and 5 mM sodium butyrate 4–6 h post-transfection. We harvested SFV or VEEV RVP-containing supernatants 2 days post-transfection, and EEEV or SINV RVP-containing supernatants 3 days post-transfection. Supernatants were then centrifuged at $3000 \times g$ for 5 min, filtered using a 0.45 μm filter, and frozen at −80 °C for storage. RVP tubes were thawed once and used thereafter.

## Expression and purification of virus-like particles

We produced EEEV virus-like particles (VLPs) using a previously described vector that encodes the structural polyprotein of EEEV strain PE6 (GenBank: AAU95735.1) with capsid mutation K67N[20]. We transfected Expi293F™ (Thermo Fisher Scientific) cells with ExpiFectamine™ 293 Transfection Kit and added enhancer 20 h after transfection according to the manufacturer's instructions. We harvested supernatant 96 h post-transfection and clarified these by centrifugation at $3000 \times g$ for 30 min. We then performed sucrose cushion ultracentrifugation by adding 5 ml of 35 % (w/v) sucrose and 5 ml of 70 % (w/v) sucrose beneath the clarified supernatant and by using a Beckman SW28Ti rotor to centrifuge samples at $110,000 \times g$ for 5 hours. We collected VLPs at the interface of the 35 % (w/v) sucrose and 70 % (w/v) sucrose cushion, then concentrated samples using a 100-kDa Amicon filter (Sigma). We loaded concentrated VLPs onto a 20–70 % (w/v) continuous sucrose gradient and centrifuged samples at $210,000 \times g$ for 1.5 h. We collected VLP bands and stored VLPs at 4 °C. VLPs were buffer exchanged into Dulbecco's Phosphate Buffered Saline (DPBS) (Thermo Fisher Scientific #: 14190-144) using a 100-kDa Amicon filter (Sigma) right before use. We confirmed particle integrity and the absence of degradation products using SDS-PAGE gel electrophoresis (see Supplementary Fig. 2d) in addition to negative stain electron microscopy. VLPs were always used within 30 days of purification.

We produced SFV VLPs by transiently transfecting HEK293T cells grown in adherent culture with a vector encoding the structural polyprotein of SFV strain SFV4 (GenBank: AKC01668.1)[3] using lipofectamine 3000 (Invitrogen) according to the manufacturer's instructions. We harvested supernatant 24 h and 48 h post-transfection and clarified these by centrifugation at $3000 \times g$ for 10 min. We then performed PEG-precipitation by mixing clarified supernatants to a final concentration of 7% (v/v) PEG 6000 and 2.3% (v/v) NaCl and incubated at 4 °C overnight. Precipitates were pelleted by centrifugation at $4000 \times g$ for 30 min and resuspended in PBS. We loaded resuspended VLPs onto a 20–70% continuous sucrose gradient and centrifuged samples at $210,000 \times g$ for 1.5 h. We collected VLP bands and buffer exchanged using a 100-kDa Amicon filter (Sigma). VLPs were buffer exchanged into DPBS (Thermo Fisher Scientific Cat#: 14190-144) using a 100-kDa Amicon filter (Sigma) right before use. We confirmed particle integrity and the absence of degradation products using SDS-PAGE gel electrophoresis (see Supplementary Fig. 2d) in addition to negative stain electron microscopy. SFV VLPs were always used within 30 days of purification.

## Generation of cell lines ectopically expressing VLDLR constructs

We designed VLDLR single LA repeat constructs and VLDLR–LDLRAD3$_{D1}$ chimeric receptor by replacing the ligand-binding domain of human VLDLR (residues 30-355, GenBank NP_003374.3) with single LA repeats of VLDLR (e.g., LA1 residues 31–69, LA2 residues 70–110, LA3 residues 111–151, LA4 residues 152–190, LA5 residues 191–231, LA6 residues 237–275, LA7 residues 276–314, LA8 residues 316–355) or LDLRAD3$_{D1}$ (residues 28-65, GenBank AAH42754.2). Constructs were tagged with a Flag tag flanked by short linkers ("EDYKDDDDKGS"; the Flag tag is underlined) so that their expression could be monitored by anti-Flag tag antibody staining. To avoid potential issues with improper signal peptide processing, rather than tagging at the N terminus, we inserted the Flag tag internally at the junction between the second EGF-like module and the β-propeller domain residues, replacing residue G438 in full-length human VLDLR. Mutants of VLDLR LA1 (W50F, W50R, D53A, D55I, D57A) or LA2 (W89F, W89R, D92A, D94I, D96A) single LA repeat constructs were generated by site-directed mutagenesis. We subcloned each of these constructs into the backbone of the lentiGuide-Puro vector (provided by F. Zhang, Addgene #52963)[45], packaged lentivirus in HEK293T cells, and used lentiviruses to generate stably transduced K562 cells through selection with 2 μg ml⁻¹ puromycin.

For immunostaining of cells expressing Flag-tagged receptors, we added an APC-conjugated rat anti-DYKDDDDK (anti-Flag) antibody (BioLegend Cat#: 637307) or an isotype control antibody (BioLegend Cat#: 402306) in binding buffer (2% [v/v] goat serum in PBS), according to the manufacturer's recommendation. Cells were washed with binding buffer, then blocked in blocking buffer for 30 min (5% [v/v] goat serum in PBS) at 4 °C. After one more wash with binding buffer, cells were incubated with the antibodies for 30 min at 4 °C. Following incubation, we washed cells twice with binding buffer, twice with chilled PBS, fixed them with 2% (v/v) formalin, and detected cell surface receptor expression by flow cytometry. For staining of cells stably transduced with lentivirus to express full-length, untagged human VLDLR[3], we added a mouse anti-human VLDLR antibody (GeneTex Cat#: GTX79552) or an isotype control antibody (BD Biosciences Cat#: 557351) to binding buffer. Cells were incubated with the antibodies for 30 min following a 30 min incubation with blocking buffer as described above and were washed three times in binding buffer afterwards. We then incubated cells with a PE-conjugated donkey anti-mouse F(ab')$_2$ fragment (Jackson ImmunoResearch Cat#: 715-116-150) for 30 min at 4 °C. Cells were washed twice with binding buffer, twice with PBS, fixed with 2% (v/v) formalin, and subjected to analysis by flow cytometry using an iQue3 Screener PLUS (Intellicyt) with IntelliCyt ForeCyt Standard Edition (version 8.1.7524) (Sartorius) software. Antibody staining was visualized using FlowJo (version 10.6.2).

## Recombinant protein production

The VLDLR$_{LBD}$-Fc fusion protein, which contains human VLDLR residues 31–355 (GenBank NP_003374.3) and human IgG1 Fc[3], was generated using transient transfection of Expi293F cells and co-expression with the chaperone receptor-associated protein RAP (full-length human RAP (residues 1–353, including the signal sequence) (GenBank NP_002328)[3] and purified from clarified culture supernatants using protein A affinity chromatography with MabSelect™ PrismA resin (Cytiva Cat#: 17549801). The resin was washed with 400 column volumes of TBS buffer (20 mM Tris, 150 mM NaCl, pH 7.5) containing 10 mM EDTA to unfold VLDLR$_{LBD}$-Fc and elute RAP. To recover RAP, the washes were collected, dialyzed into TBS using a Slide-A-Lyzer™ Dialysis Cassette (10 K MWCO) (Thermo Fisher Scientific Cat#: 66810), and concentrated. VLDLR$_{LBD}$-Fc was refolded on the column by passing 200 column volumes of TBS buffer containing 2 mM CaCl$_2$ onto the column. VLDLR$_{LBD}$-Fc was subsequently eluted according to the manufacturer's recommendations and buffer exchanged into TBS containing 2 mM CaCl$_2$ by using a Slide-A-Lyzer™ Dialysis Cassette (10 K MWCO). Both RAP and VLDLR$_{LBD}$-Fc were passed through a size-exclusion chromatography column (Superdex 200™ Increase) (Cytiva) with peak fractions pooled and stored. RAP was stored in TBS buffer and VLDLR$_{LBD}$-Fc was stored in TBS buffer containing 2 mM CaCl$_2$.

The MXRA8$_{ect}$-Fc protein, which contains human MXRA8 residues 20–337 (GenBank NP_001269511.1) and human IgG1 Fc[3], was expressed in a vector and purified from the supernatant of transiently transfected Expi293F cells. The VLDLR$_{LA1}$-Fc fusion protein, which contains a TPA signal peptide followed by VLDLR residues 31–69 and human IgG1 Fc[46], was cloned into a pVRC vector and expressed using transient transfection of Expi293F cells and co-expressed with full-length human RAP. Culture supernatants were clarified followed by protein A affinity chromatography with MabSelect™ PrismA resin using the manufacturer's protocol. MXRA8$_{ect}$-Fc and VLDLR$_{LA1}$-Fc were eluted from the column by following the manufacturer's protocol, buffer exchanged into TBS and TBS containing 2 mM CaCl$_2$ respectively, using a Slide-A-Lyzer™ Dialysis Cassette (10 K MWCO), and passed on a size exclusion chromatography column (Superdex 200™ Increase) (Cytiva) with peak fractions pooled and stored. MXRA8$_{ect}$-Fc was stored in PBS and VLDLR$_{LA1}$-Fc was stored in TBS supplemented with 2 mM CaCl$_2$.

## Cell surface Fc fusion protein binding assays

We generated EEEV FL91-469 E2 K156A and R157A, as well as SFV E1 K345A and K347A, mutant constructs on the background of wild-type EEEV E3–E2–(6K/TF)–E1 (FL91-469 strain, GenBank: Q4QXJ7.1), or wild-type SFV E3–E2–(6K/TF)–E1 (SFV4 strain, GenBank: AKC01668.1) sequence-expressing pCAGGS vectors using site-directed mutagenesis. We transfected HEK293T cells with pCAGGS alphavirus E3–E2–(6K/TF)–E1 wild-type or mutant expressor plasmids using Lipofectamine 3000 (Invitrogen) or an empty pCAGGS vector. We detached cells 48 h post-transfection with TrypLE™ Express (Thermo Fisher Scientific) and washed them in binding buffer (20 mM Tris pH 7.5, 150 mM NaCl, 2 mM CaCl$_2$, 2% [v/v] goat serum) followed by incubation in blocking buffer (20 mM Tris pH 7.5, 150 mM NaCl, 2 mM CaCl$_2$, 5% [v/v] goat serum). Cells were then incubated with increasing concentrations of MXRA8$_{ect}$-Fc or VLDLR$_{LBD}$-Fc fusion protein in binding buffer with 2% (v/v) goat serum for 1 h at 4 °C. We then washed cells three times in binding buffer and incubated them with a $R$-phycoerythrin (PE)-coupled goat anti-human F(ab')2 fragment (Jackson ImmunoResearch Cat#: 109-116-098) in binding buffer for 30 min at 4 °C. We washed cells three times with binding buffer, then twice with binding buffer without goat serum, and fixed cells with 2% (v/v) formalin. We measured cell binding by monitoring PE-intensity and percent positivity by flow cytometry. To monitor expression of mutant EEEV E3–E2–(6 K/TF)–E1, we also stained cells expressing wild-type and mutant EEEV glycoproteins with mouse anti-EEEV E1 antibody clone 1A4B.6 (Sigma-Aldrich Cat#: MAB8754) or a mouse IgG2b isotype control (BD Biosciences Cat#: 557351). To monitor expression of mutant SFV E3–E2–(6 K/TF)–E1, we stained cells expressing wild-type and mutant SFV glycoproteins with mouse anti-SFV immune ascitic fluid (ATCC Cat#: VR-1247AF) or a control mouse IgG2b antibody (BD Biosciences Cat#: 557351). Staining of wild-type and mutant EEEV or SFV E3–E2–(6 K/TF)–E1 was detected with a PE-conjugated donkey anti-mouse F(ab')$_2$ fragment (Jackson ImmunoResearch Cat#: 715-116-150) using flow cytometry. An example of the flow cytometry gating scheme used to quantify Fc fusion protein cell surface immunostaining is provided in Supplementary Fig. 7d.

## Reporter virus particle infection assays

We incubated transduced K562 cells with RVPs in the presence of 5 µg ml$^{-1}$ polybrene for experiments shown in Fig. 2c, f and Supplementary Fig. 1d–g, and without polybrene for experiments shown in Fig. 3d, e. Twenty-four hours post-infection, cells were harvested, washed twice with phosphate buffered saline (PBS), and fixed in PBS containing 2% (v/v) formalin. GFP expression was measured by flow cytometry using an iQue3 Screener PLUS (Intellicyt) with IntelliCyt ForeCyt Standard Edition version 8.1.7524 (Sartorius) software. An example of the flow cytometry gating scheme used to quantify GFP-

expression after infection with EEEV RVPs is provided in Supplementary Fig. 1c.

For experiments in which infection levels by wild-type and mutant RVPs were compared, wild-type and mutant EEEV RVPs were first titered on Vero E6 cells seeded in a 96-well plate using a serial tenfold dilution of the RVP stocks. At 24 h post-infection, numbers of GFP-positives cells were counted using a fluorescence microscope and used to calculate RVP titers as infectious unit per milliliter (IU ml$^{-1}$), assuming that at high dilution factors, 1 GFP-positive cell = 1 infectious unit (IU), given that RVPs can only infect cells for one cycle. Representative titers are shown in Supplementary Fig. 7f. Wild-type and mutant SFV RVPs were titered on C6/36 (*Aedes albopictus* cells), because SFV depends on VLDLR to infect Vero E6 cells[3]. Wild-type and mutant EEEV FL91-469 RVPs were used at an MOI of 10 (Fig. 3d). Wild-type and mutant EEEV PE6 RVPs and SFV RVPs were used at an MOI of 2 (Fig. 3e and Supplementary Fig. 7g).

## Murine cortical neuron culture and infection

Primary murine cortical neurons were obtained commercially (Thermo Fisher Scientific A15586). Neurons were thawed and cultured according to the manufacturer's protocol. In brief, neurons were thawed and assessed for viability. Upon confirmation of a viability >90%, we seeded neurons in 96 well plates coated with 4.5 µg cm$^{-2}$ poly-D-lysine at $4 \times 10^4$ neurons per well. Neurons were maintained in complete medium (Neurobasal medium (Thermo Fisher Scientific #21103049) supplemented with 2% (v/v) B27 supplement (Thermo Fisher Scientific #17504) and 0.5 mM GlutaMAX-I (Thermo Fisher Scientific #35050)). We performed a half medium change the next day with pre-warmed complete medium. After culturing for 3 days, neurons were infected with GFP-expressing WT and mutant RVPs for EEEV FL91-469 at MOI of 2. Cells were imaged every 4 h for 24 h using the Incucyte S3 Live Cell Imaging system (Sartorius) with Incucyte S3 Software (version 2023B) (Sartorius) using a 20X objective. GFP-positive neurons were scored as cells with a threshold signal greater than 2 green calibrated units above background, using a Top-hat background subtraction method. Twenty-four hours post-infection, we calculated percent GFP-positive cells by dividing the area of GFP signal above background by the total area covered by neuronal cell bodies multiplied by 100. Relative infection was calculated as follows: Relative infection (%) = (% GFP positive cells infected by the EEEV FL91-469 RVP of interest/% GFP-positive cells infected by WT EEEV FL91-469 RVPs) × 100.

## EEEV VLP:VLDLRLBD-Fc complex cryo-EM structure determination

We mixed 2 µl of EEEV VLP at a concentration of 1.6 mg ml$^{-1}$ in DPBS (Thermo Fisher Scientific Cat#: 14190-144) with 2 µl of VLDLR$_{LBD}$-Fc in TBS supplemented with 2 mM CaCl$_2$ at a concentration of 1.6 mg ml$^{-1}$. We then immediately applied the mixture to glow-discharged Quantifoil grids (R 0.6/1, 300 mesh, gold, Electron Microscopy Sciences [EMS] Cat#: Q350AR-06) and blotted once for 6 s after a wait time of 30 s in 100% humidity at 4 °C. Grids were then plunged into liquid ethane using an FEI Vitrobot Mark IV (Thermo Fisher Scientific). We collected an initial cryo-EM dataset on an FEI Titan Krios microscope (300 kV) (Thermo Fisher Scientific) at the Harvard Cryo-Electron Microscopy Center. Movies were recorded using a K3 detector (Gatan) with a defocus range of −1.0 to −2.5 µm. Automated single-particle data acquisition was performed with SerialEM, with a nominal magnification of ×81,000 in counting mode, which yielded a calibrated pixel size of 1.06 Å. Raw movies were motion-corrected using MotionCor2 (version 1.6.4)[47] and combined into micrographs, and the defocus value for each micrograph was determined using CTFFIND-4.1 (version 4.1.14)[48]. A total of 18,493 particles were boxed using crYOLO (version 1.8.2)[49] from 3,915 micrographs. Chosen particles were extracted from micrographs and binned two times (pixel size 2.12 Å) in RELION (version 3.1.4)[50]. 2D classification was performed to discard bad particles, a

total of 16,143 particles from good class averages were selected for 3D classification, and 10,628 particles were selected for reconstruction of the entire VLP. With I3 symmetry, the resolution of the VLP map was 4.3 Å.

We next used the block-based reconstruction approach to increase the resolution of maps[12]; 637,680 blocks from the q3 spikes were subjected to 3D auto-refinement without alignment in RELION (version 3.1.4)[50]. With C1 symmetry, we obtained a 3.1 Å map, which was further processed using DeepEMhancer (version 20210511)[51]. Density for the viral glycoproteins, associated N-linked glycans, and capsid was high in quality (Supplementary Fig. 4a). We could also observe weak density consistent in shape and size with a single VLDLR LA repeat wedged in clefts formed by adjacent E2–E1 protomers but only at low map contour levels in maps that had not been post-processed by DeepEMhancer[51], suggesting that LA repeats bound the viral glyco-protein weakly and with partial occupancy.

To improve the quality of LA repeat density in the cleft, we devised a strategy to collect a second, larger cryo-EM dataset on the same instrument, with the goal of separating unliganded from liganded spikes. For this second dataset, 82,189 VLPs were boxed from 9963 micrographs using crYOLO (version 1.8.2)[49], and 2D classification was performed to discard bad particles. A total of 63,790 particles in good class averages were selected for 3D classification, and 19,954 particles were selected for VLP reconstruction. With I3 symmetry, the resolution of the EEEV VLP:VLDLR$_{LBD}$-Fc complex map for the second dataset was 4.9 Å. We then combined VLPs from both datasets, and the resolution of the combined EEEV VLP:VLDLR$_{LBD}$-Fc complex map was 4.4 Å. We next again performed block-based reconstruction of q3 spikes, with 3D classification performed without alignment to discard particles that did not contain density for bound LA repeats, suggesting that they represented unliganded spikes. 185,420 blocks thought to represent liganded spikes were selected for map reconstruction. With C1 symmetry, the resolution of the map for these spikes was 3.5 Å. CTF refinement was performed, and the resolution of the map was improved to 3.3 Å. To improve features for the LA repeat density, masked refinement was performed over the E2–E1 cleft that contained the strongest density for the LA repeat. The resolution of masked refinement map is 3.9 Å, and RELION (version 3.1.4) post-processing yielded a 3.5 Å with improved density and clear features that allowed unambiguous tracing of the polypeptide backbone and placement of residues with bulky side chains. The DeepEMhancer-processed (version 20210511)[51] 3.1 Å map in which the details of the spike protein and capsid are best resolved, and a separate 3.5 Å VLDLR-focused map in which the LA repeat density is resolved, were deposited.

## SFV VLP:VLDLRLBD-Fc cryo-EM structure determination

We mixed 2 µl of SFV VLP at 3.3 mg ml⁻¹ in DPBS with 2 µl of VLDLR$_{LBD}$-Fc protein at 3.3 mg ml⁻¹ in TBS supplemented with 2 mM CaCl₂. The sample was then immediately applied to glow-discharged Quantifoil grids (R 1.2/1.3, 400 mesh, copper grids, EMS Cat#: Q450CR1.3), blot-ted once for 6 s after a wait time of 30 s in 100% humidity at 4 °C, and plunged into liquid ethane using an FEI Vitrobot Mark IV (Thermo Fisher Scientific). Cryo-EM datasets were collected on an FEI Titan Krios microscope (300 kV) (Thermo Fisher Scientific) at the Harvard Cryo-Electron Microscopy Center. Movies were recorded using a K3 detector (Gatan) with a defocus range of −1.5 to −2.5 µm. Automated single-particle data acquisition was performed with SerialEM, with a nominal magnification of 105,000 X in counting mode, which yielded a calibrated pixel size of 0.825 Å. Raw movies were motion-corrected using MotionCor2 (version 1.6.4)[47] and combined into micrographs, and the defocus value for each micrograph was determined using CTFFIND-4.1 (version 4.1.14)[48]. A total of 114,592 particles were picked from 5738 micrographs using crYOLO (version 1.8.2)[49], and particles were extracted from micrographs and binned two times (pixel size 1.65 Å) in RELION (version 3.1.4)[50]. Two rounds of 2D classification were

performed to discard bad particles, and 43,883 particles in good class averages were selected for 3D classification. A total of 37,726 particles were selected for VLP reconstruction. With I3 symmetry imposed, the resolution of the SFV VLP:VLDLR$_{LBD}$-Fc cryo-EM map was 4.9 Å. VLP maps showed features consistent with single LA repeats bound near the base of E1 at the two- and five-fold symmetry axes, but the reso-lution was too poor for unambiguous placement of LA repeats. To improve the resolution and map quality of the region and to allow for model building, we performed a block-based reconstruction[12] at the icosahedral five-fold axis of the VLP. We performed 3D classification without alignment to the discard blocks that did not have density with clear features for LA repeats, and 439,486 blocks were selected for final reconstruction. We performed 3D auto-refinement in RELION (version 3.1.4)[50]. With C1 symmetry, the final resolution of the five-fold axis block-based reconstruction map was 3.7 Å. To improve features for the LA repeat density, masked refinement was performed over the E1 region that contained strongest density for the LA repeats, which was the five-fold axis. The resolution of the map after masked refine-ment was 3.8 Å, and we used a separate map generated using Dee-pEMhancer (version 20210511)[51] to guide LA repeat placement and model building. The five-fold block (3.7 Å) and five-fold-focused block (3.8 Å) maps were deposited.

## EEEV virus-like particle cryo-EM structure determination

We applied 3 µl of purified EEEV VLP, from structural polyprotein sequence derived from EEEV strain PE6 (GenBank: AAU95735.1)[20], at a concentration of 3.2 mg ml⁻¹ in DPBS (Thermo Fisher Scientific Cat#: 14190-144) onto glow-discharged Quantifoil grids (R 0.6/1 300 mesh, gold, EMS Cat#: Q350AR-06), blotted once for 5 s after a wait time of 15 s at 4 °C in 100% humidity, and plunged into liquid ethane using an FEI Vitrobot Mark IV (Thermo Fisher Scientific). Cryo-EM datasets were collected using a FEI Titan Krios microscope (300 kV) (Thermo Fisher Scientific) equipped with a K3 detector (Gatan) at the Harvard Cryo-Electron Microscopy Center. Movies were recorded at a defocus range of −1.0 to −2.5 µm. Automated single-particle data acquisition was performed with SerialEM, with a magnification of ×81,000 in counting mode, which yielded a calibrated pixel size of 1.06 Å. Raw movies were corrected for beam-induced motion using MotionCor2 (version 1.6.4)[47]. The defocus values were estimated by CTFFIND-4.1 (version 4.1.14)[48]. A total of 42,043 particles were picked using crYOLO (version 1.8.2)[49], and particles were extracted from 3753 micrographs and bin-ned two times (pixel size 2.12 Å) in RELION (version 3.1.4)[50]. Extracted particles were then subjected to two rounds of reference-free 2D classification using RELION (version 3.1.4)[50]. A total of 36,975 particles were selected from good 2D classes and subjected to 3D classification with icosahedral symmetry, and these particles were classified into five classes using the 4.3 Å map of EEEV VLP:VLDLR$_{LBD}$-Fc complex VLP as the initial model. A total of 36,975 particles were subjected to 3D auto-refinement and postprocessing, yielding a 4.6 Å reconstruction of the EEEV VLP. To improve the resolution of the map, we used the block-based reconstruction method[12] centered on the q3 spikes. A total of 1,802,280 blocks were extracted without binning and subjected to 3D classifications. A total of 432,897 particles were selected for initial refinement, which was followed by CTF refinement, and a final round of refinement that generated a 3.1 Å map of the q3 spikes. We then used masked refinement to obtain a 3.0 Å density map of the asymmetric unit, which includes four copies of E3–E2–E1 heterotrimers and four copies of the capsid. This map was processed using DeepEMhancer (version 20210511)[51] for cryo-EM volume post-processing to generate a map for model building.

## Model building

To generate an initial model of VLDLR$_{LBD}$-Fc-bound EEEV spike pro-teins, we docked the AlphaFold2-predicted[52] EEEV E3, E2, E1, and cap-sid structures into a DeepEMhancer (version 20210511)[51] post-

processed map (3.1 Å) (Supplementary Fig. 3). We generated an AlphaFold2[52] model of VLDLR LA1, and docked this LA repeat as a rigid body into the EEEV E2–E1 cleft with the strongest LA repeat density into maps that had been focused on this particular cleft (3.5 Å) (Supplementary Fig. 3). This was followed by iterative model building in *Coot* (version 0.9.8.8)[53]. PHENIX (version 1.21-5207)[54] was used to refine the model by iterative positional and B-factor refinement in real space. Deposited atomic coordinates include one copy of an LA repeat bound in the asymmetric unit, which includes four copies of E3–E2–E1 heterotrimers.

To generate an initial model of EEEV unliganded spike proteins, we docked the AlphaFold2-predicted[52] EEEV E3, E2, E1, and capsid structures into the DeepEMhancer (version 20210511)[51] post-processed map. This was followed by iterative model building in *Coot* (version 0.9.8.8)[53], and PHENIX (version 1.21-5207)[54] was used to refine the model by iterative positional and B-factor refinement in real space.

To generate an initial model of VLDLR$_{LBD}$-Fc-bound SFV spike proteins, we docked AlphaFold2-predicted[52] SFV E3, E2, E1, and capsid structures into the DeepEMhancer (version 20210511)[51] post-processed map (3.7 Å). A combination of post-processed maps and masked maps focused on the VLDLR density (3.8 Å) for one of the bound LA repeats was used to dock an AlphaFold2[52] model of VLDLR LA2 as a rigid body. This was followed by iterative model building in *Coot* (version 0.9.8.8)[53], and PHENIX (version 1.21-5207)[54] was used to refine the model by iterative positional and B-factor refinement in real space. Deposited atomic coordinates include one copy of an LA repeat bound in the asymmetric unit, which includes four copies of E3–E2–E1 heterotrimers. All initial docking steps into maps were performed using UCSF Chimera (version 1.17.1)[55]. Software used in this project was curated by SBGrid[56]. Data collection and refinement statistics are provided in Supplementary Table 1. PyMOL (version 2.5.5) and UCSF Chimera (version 1.17.1)[55] were used to generate figures. ResMap (version 1.1.4)[57] was used to provide local resolution estimates for cryo-EM maps.

## Biolayer interferometry binding assays

We performed biolayer interferometry experiments with an Octet RED96e (Sartorius) and analyzed data using ForteBio Data Analysis HT version 12.0.1.55 software. For the determination of binding affinities, an anti-EEEV E1 mouse monoclonal antibody (clone 1A4B.6 Millipore Cat#: MAB8754MI) was loaded onto Anti-Mouse IgG Fc Capture (AMC) Biosensors (Sartorius Cat#: 18-5088) at a concentration of 250 nM in kinetic buffer (TBS supplemented with 2 mM $CaCl_2$ and 0.1% [w/v] BSA) for 600 s. After a baseline measurement for 60 s in kinetic buffer, EEEV VLPs (prepared as noted above) at a concentration of 1 μM were loaded onto sensor tip surfaces for 1 h. The signal was then allowed to equilibrate for an additional 1 h in kinetic buffer. An additional baseline measurement for 60 s in kinetic buffer was performed. For the association phase, tips were dipped into solution containing serial dilutions of VLDLR$_{LBD}$-Fc (concentration range of 6.25 nM to 0.53 nM), VLDLR$_{LA1}$-Fc (concentration range of 10 μM to 0.31 μM), or MXRA8$_{ect}$-Fc (concentration range of 10 μM to 0.31 μM) in kinetic buffer for 300 s. Finally, a 300 s disassociation in kinetic buffer was performed. A 1:1 model was used for fitting the data. Scatchard plot analysis was used to analyze the EEEV VLP-VLDLR$_{LA1}$-Fc given the fast off rate.

For experiments with EDTA, anti-EEEV E1 mouse monoclonal antibody (clone 1A4B.6, Millipore Cat#: MAB8754MI) was loaded onto anti-mouse IgG Fc Capture Biosensors at 250 nM in kinetic buffer, and EEEV or CHIKV VLPs at a concentration of 1 μM were loaded onto sensor tips. For SFV VLPs, we loaded SKT05 monoclonal antibody (humanized IgG, provided as a gift by M. Sutton and M. Roederer)[58] onto anti-human IgG Fc Capture Biosensors at 450 nM in kinetic buffer. During the association phase, tips were dipped into solutions containing 20 nM VLDLR$_{LBD}$-Fc, either alone, with 10 mM EDTA, or with 100 μg ml$^{-1}$ (2.6 μM) RAP during the association phase of the

experiment. 20 nM MXRA8$_{ect}$-Fc was additionally used in experiments with CHIKV VLPs.

## Statistical analysis

Data were deemed statistically significant when $p$ values were <0.05 using version 10.0.3 of GraphPad Prism. Experiments were analyzed by one-way or two-way ANOVA with multiple comparison correction.

## Reporting summary

Further information on research design is available in the Nature Portfolio Reporting Summary linked to this article.

## Data availability

Structural coordinates and associated maps are available under 8UA4 (EEEV VLP:VLDLR$_{LBD}$-Fc complex), EMD-42050 (EEEV VLP:VLDLR$_{LBD}$-Fc complex), 8UA9 (EEEV VLP alone), EMD-42055 (EEEV VLP alone), 8UA8 (SFV VLP:VLDLR$_{LBD}$-Fc complex), and EMD-42054 (SFV VLP:VLDLR$_{LBD}$-Fc complex). Source data are provided with this paper.

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

## Acknowledgements

J.A. is a recipient of Burroughs Wellcome Fund Investigators in the Pathogenesis of Infectious Disease Award. This work was also supported by a Vallee Scholar Award (J.A.), Smith Family Foundation Odyssey Award (J.A.), Charles E.W. Grinnell Trust Award (J.A.), NIH grant R01-AI182377 (J.A.), T32AI700245 (C.M.), T32GM144273 (H.V. and K.N.S.), T32GM008313 (H.V.), and in part by a grant to Harvard Medical School from the Howard Hughes Medical Institute

through the James H. Gilliam Fellowships for Advanced Study program (L.E.C.). The authors also acknowledge S. Jenni for cryo-EM data review and discussion and thank the staff at the Harvard Cryo-EM Center for Structural Biology at Harvard Medical School, including S. Sterling, R. Walsh, S. Rawson, M. Mayer, and R. Nair, for help with data acquisition, pre-processing, and storage.

## Author contributions

P.Y. produced EEEV and SFV VLPs, produced recombinant Fc fusion proteins, performed biolayer interferometry experiments, performed immunostaining experiments, and determined the cryo-EM structures of the EEEV VLP:VLDLRL$_{BD}$-Fc and SFV VLP:VLDLR$_{LBD}$-Fc complexes. W.L. generated cell lines with wild-type and mutant receptors, generated RVPs, produced recombinant proteins, performed immunostaining experiments, and conducted infectivity studies with RVPs. X.F. determined the cryo-EM structure of unliganded EEEV VLPs. J.P. helped build and refine atomic models. C.M., H.V., L.E.C. and K.N.S. helped produce recombinant Fc fusion proteins. S.A.C. and A.C. produced VLPs, and V.B. generated RVPs and VLPs. H.B. collected and analyzed data. J.A. acquired funding. P.Y., W.L. and J.A. wrote the original draft of the manuscript, P.Y., W.L., X.F. and J.A. generated figures, and all authors participated in reviewing and editing the manuscript.

## Competing interests

The authors declare no competing interests.
