## [Peer Review File · Nature Communications]

Structural basis for VLDLR recognition by eastern equine encephalitis virusREVIEWER COMMENTS

Reviewer #1 (Remarks to the Author):

This manuscript by Abraham et al. describes structural studies of EEEV binding to the VLDL receptor. This follows on the identification of VLDLR and the structurally related ApoER2 as a receptor for EEEV, SFV and SINV by Abraham and colleagues. This paper defines the structure of an LA repeat bound to EEEV or SFV, detailing their two distinct binding sites on the E2 protein vs E1 domain III. The authors then confirm the relevance of the binding interactions by mutagenesis and functional studies of the receptor and the viral envelope proteins. The paper is very well-written and the data are clear and the interpretations thoughtful. I think the paper will be an excellent addition to our overall understanding of virus interactions with LDLR family members, as well as an important contribution to the alphavirus receptor field.

The following points should be addressed.

Major points:

1. Recent structural data describes the binding of avian MXRA8 to SINV and other alphaviruses with avian reservoirs (doi: 10.1016/j.cell.2023.09.007). Given that this paper is now published (although it was not when the authors submitted), it would be helpful if the authors could comment on this use of a very different receptor by SINV, vs. SINV's use of VLDLR. Structural comparisons of binding interactions would be helpful. Do the authors have any insights into which of these receptors is more important for SINV?
2. There are older data indicating that SFV receptor binding is not calcium dependent. Those studies did not use cells shown to be strongly dependent on expression of VLDLR, so HSPG binding may have played an important role. Can the authors comment on whether SFV or EEEV binding to VLDLR is inhibited by calcium chelation in any of their systems?
3. I thought that the last paragraph of the discussion raised very interesting points. It would be helpful to the general reader to explain what is meant by the open vs. closed conformation of the LDLR family ligand binding domain.
4. Can the authors please comment on the technical differences that enabled them to obtain the structure of EEEV VLP with the VLSLR LA domain while the Cao paper reported

that they could not? Maybe I missed this in the manuscript, but if not it would be helpful to point this out for the field.

Minor points:

1. I was puzzled by the inclusion of polybrene in the reporter virus entry studies. Please comment on this and whether it was required.
2. Please specify the cut-off for positive in Fig. 1b.
3. Please define cyan highlights in Fig. 2a.

Reviewer #2 (Remarks to the Author):

The manuscript from Yang et al reports thorough, comparative structure-function studies of VLDLR recognition by EEEV and SFV. Cell-surface binding studies are used to identify the specific VLDLR LA repeats that EEEV and SFV bind. Cryo-EM structures are determined of EEEV and SFV VLPs in complex with soluble receptor constructs. Interestingly, EEEV and SFV have related but different modes of receptor recognition: VLDLR interacts with two basic residues on the EEEV E2 subunit whereas it interacts with two basic residues on the SFV E1 subunit. The general mode of binding is conserved among viral proteins and endogenous ligands that interact with LDLR-related proteins. The authors validate their structural findings with site-directed mutagenesis studies, which are well controlled. The results add to the growing body of knowledge concerning the interactions of viral and host proteins with LDLR-related proteins, with implications for viral evolution and adaptation.

Overall this is a very well written and presented manuscript with findings that will be of broad interest to virologists and structural biologists. The experiments are performed to a high standard, and the conclusions are justified by the data. The authors should, however, show one or more supplementary figures/panels that display the EM map and model of each interface. The resolutions of the complexes are in the low-to-mid 3Å, and as the authors note they contain some heterogeneity due to averaging of different LA repeats. Thus, it is difficult to determine the confidence of side chains and hydrogen bonds/salt

bridges that are shown in various figures (2b,e; 3a). There is some attempt to show this in Extended Data Fig 3 and 9, but the images are small and it is difficult to tell whether sidechains are within the map. Some density for the EM maps is shown in Extended Data Fig 4, but these are not interface residues, so the value of showing this is rather limited.

Reviewer #3 (Remarks to the Author):

Comments for Authors

Semliki forest virus (SFV) and eastern equine encephalitis virus (EEEV) are two alphaviruses that can cause disease in humans. EEEV is of special clinical concern due to its high case fatality rate, the risk of neurological sequelae, and the lack of approved therapeutics or vaccines. In this manuscript, Yang and Li et al. report the structural basis for binding of SFV and EEEV to their cellular receptor VLDLR. Using single-particle cryo-electron microscopy the authors identified two separate binding modes for EEEV and SFV with the ligand-binding domain of VLDLR: EEEV engages VLDLR via the viral glycoprotein E2; SFV via the viral fusion protein E1. The structural observations are functionally validated in cellulo, using a structurally informed set of virus and VLDLR mutant constructs. Importantly, while cryo-EM structures of other alphavirus-receptor pairs are already available, they cannot be extrapolated to SFV and EEEV. This scientifically well-designed study therefore contributes a novel structural explanation of differential receptor-usage in the family of alphaviruses.

Major comments

1. The structural data shows that the calcium coordinating amino acids in the LA repeats are essential for interaction with both EEEV and SFV. SFV cell entry is not dependent on calcium and proceeds unperturbed even in the presence of a calcium chelator (e.g. Dube et al. 2016) – do the authors have any information on the effect of calcium chelators on VLDLR?

Minor comments

1. Line 79-80: reference mix-up? Heidner et al. 1996 (Reference 2) is cited for equine infection with encephalitic alphaviruses in the Americas; data is however found in Corrin et al. 2021 (Reference 4).
2. Line 93-94: wording. “Importantly, while LDLRAD3 is a receptor for VEEV, it is not a receptor for any other [tested] alphavirus.”

Reviewer #4 (Remarks to the Author):

Previously, L. Clark and colleagues described very-low density lipoprotein receptor (VLDLR) as a receptor for certain alphaviruses such as eastern equine encephalitis virus (EEEV) and Semliki forest virus (SFV). It was shown that the VLDLR ligand-binding domain (LBD), which contains LDLR class A (LA) repeats, is needed to support alphavirus infection and that the viral envelope glycoproteins (E proteins) engage with the receptor.

In this manuscript, Yang et al. determine the structural basis for binding of EEEV and SFV to VLDLR by cryo-electron microscopy (cryo-EM). While VLDLR contact sites between EEEV and SFV are divergent, the authors show that basic residues in the viral E proteins are required for interaction with the receptor. Using mutated E proteins and VLDLR LA repeat constructs, importance of contact sites in VLDLR and E proteins determined by cryo-EM was further validated by cell surface staining experiments and through infectivity assays.

Finally, the authors compared the identified binding mode to the interactions of LA repeats with other viruses and ligands.

In sum, the study gives important insights into the interaction of alphaviruses with lipoprotein receptors. The study is novel and original. References are appropriately cited. Overall, the manuscript is well written. Experiments are well explained and the technology used is highly innovative and suitable for the questions addressed. However, certain questions remain that need to be discussed before publishing the manuscript. Additionally, some changes have to be implemented into the manuscript. These are listed below.

Major remarks:

1. Statistics: In Fig. 2c, d, f, g and Extended Data Fig. 1d-g it is stated that two to three biological experiments were performed. However, it seems that values of technical replicates were used to perform statistical analyses (e.g. indicated by stating that $n=9$ in Extended Data Fig. 1 when only three experiments with three technical replicates were performed). Please re-analyse the according data set and state $n=x$ correctly.
2. Extended Data Fig. 1d & Extended Data Fig. 2c & d: K652 cells expressing VLDLR LA1 seemed to be highly permissive to EEEV RVP, whereas biolayer interferometry experiments indicated low affinity of VLDLRLA1-Fc binding to EEEV VLPs. Please discuss possible explanations for these observations.

Minor remarks:

1. A VLDLRLBD-Fc construct is used throughout the study. It would be nice to illustrate the structure of the construct as done in Fig. 1a for other constructs.
2. Color code: Please make sure to explain the color code used in figures; especially colors used in alignments were not explained properly.
3. Extended Data Fig. 1: The expression of LA2Flag in K562 seemed to be lower than the expression of other constructs (see Extended Data Fig. 1a). Please comment if this could have an influence on infection with tested RVPs (e.g. VEEV RVPs did not seem to be able to engage with LA2 under tested conditions).
4. The authors show that only small conformational changes occur in the EEEV glycoprotein upon VLDLR binding. Additionally, they mention that “some LDLR-family receptors ... favor a closed conformation at low pH”. Would it be possible to modulate changes in viral E proteins and VLDLR at low pH as compared to neutral pH (e.g. by computational modulation)?
5. Extended Data Fig. 5: Some residues that are mentioned in the results part are not shown in the figure (e.g. H155). Please add them.
6. Page 9: The authors state that “...EEEV can engage LA repeats that contain either a tryptophan or a phenylalanine as the key aromatic residue. However, the LA2 W89F mutation abolished SFV RVP entry into K562 cells (Fig. 2f). These findings suggest that the polar contact the SFV E1 D327 side chain makes with the indole nitrogen of the LA repeat tryptophan (Fig. 2e) is a critical interaction and highlight a strict requirement of a

tryptophan as the LA repeat aromatic residue for engagement of the SFV spike protein.“. Wouldn't it be theoretically possible that another aromatic residue than phenylalanine could also support SFV infection? If this is the case, please re-phrase the sentence.

7. The authors should consider adding parts of their supplementary figures to the main text (e.g. Extended Data Fig. 1d-g and Extended Data Fig. 2 c).

8. Performing infection assays with EEEV and SFV harboring E proteins with mutated VLDLR contact sites could further support the findings of this manuscript.

9. Extended Data Table 1: The EEEV VLP:VLDLRLBD-Fc defocus range given in the table differs from the one given in the methods part. Please correct.

10. Please correct grammatical errors and spelling mistakes; a few times the font size/spacing changes, in Extended Data Figure 1c the labeling is shifted.

REVIEWER COMMENTS

Reviewer #1 (Remarks to the Author):

This manuscript by Abraham et al. describes structural studies of EEEV binding to the VLDL receptor. This follows on the identification of VLDLR and the structurally related ApoER2 as a receptor for EEEV, SFV and SINV by Abraham and colleagues. This paper defines the structure of an LA repeat bound to EEEV or SFV, detailing their two distinct binding sites on the E2 protein vs E1 domain III. The authors then confirm the relevance of the binding interactions by mutagenesis and functional studies of the receptor and the viral envelope proteins. The paper is very well-written and the data are clear and the interpretations thoughtful. I think the paper will be an excellent addition to our overall understanding of virus interactions with LDLR family members, as well as an important contribution to the alphavirus receptor field.

The following points should be addressed.

We thank the reviewer for their overall positive assessment of the manuscript. We provide answers to the points noted below, and in some cases, when multiple points were raised, we have broken down our answers.

Major points:

1. Recent structural data describes the binding of avian MXRA8 to SINV and other alphaviruses with avian reservoirs (doi: 10.1016/j.cell.2023.09.007). Given that this paper is now published (although it was not when the authors submitted), it would be helpful if the authors could comment on this use of a very different receptor by SINV, vs. SINV's use of VLDLR. Structural comparisons of binding interactions would be helpful.

We thank the reviewer for bringing up this important point about a recent publication that should be discussed in our manuscript. Zimmerman *et al.* recently reported that alphaviruses belonging to the western equine encephalitis virus (WEEV) complex including SINV can recognize avian orthologs of MXRA8 as cellular receptors. We agree that structural comparison of binding interactions of SINV with VLDLR vs. MXRA8 would be helpful; however, the structural analysis reported in Zimmerman *et al.* was only performed with WEEV but not with SINV. Additionally, there is no structural information available for how SINV binds VLDLR. Interestingly, SINV likely binds VLDLR using unique binding sites EEEV E2 or SFV E1 domain III binding sites because the critical basic residues that bind VLDLR in EEEV E2 (e.g., K156 and R157) or SFV E1 domain III (K345 and K347) are not conserved in SINV E2 (see Supplementary Figure 8a). We would thus not be able to provide a useful structural comparison of SINV/MXRA8 and SINV/VLDLR interaction modes.

To clarify the point that SINV likely has a distinct and unknown binding mode for VLDLR, we have added the following text to the manuscript (lines 434–443):

"In addition to serving as receptors for EEEV and SFV, VLDLR can also serve as a receptor for SINV¹. The structures suggest that SINV will use a VLDLR binding mode that is distinct from either EEEV or SFV. EEEV E2 K156 and R157, the site 1 basic residues that are critical to the interaction of EEEV with VLDLR LA repeats, are respectively replaced by a leucine and a lysine in SINV (Supplementary Fig. 8a). SFV E1 domain III residues K345 and K347, which are critical to the interaction of SFV with LA repeats, are respectively replaced by a histidine and

a leucine in SINV E2 (Supplementary Fig. 8b). Because our functional analysis suggests that at least two spike protein basic residues are required to productively engage VLDLR LA repeats, and neither site in the SINV spike protein would meet this requirement, the SINV spike protein VLDLR binding mode is likely distinct than that of the EEEV and SFV spike proteins.”

Do the authors have any insights into which of these receptors is more important for SINV?

We unfortunately do not yet have insight into whether VLDLR or MXRA8 are more important receptors for SINV. We believe, however, that the answer to this question may depend on the host that is infected and the tissue that is being infected as well (e.g., we suspect that VLDLR/ApoER2 will differ in their tissue expression patterns). For example, we know that human VLDLR and ApoER2 can serve as SINV receptors, but probably in a manner that is less efficient than their roles as receptors for SFV and EEEV based on the strengths of phenotypes we observed in our cell-based and biochemical assays (Clark *et al.* 2022, PMID: 34929721). We also reported in that prior publication that an avian ortholog of VLDLR does not serve as a receptor for SINV, but that an avian ortholog of ApoER2 does. On the other hand, Zimmerman *et al.* found that avian MXRA8 but not human MXRA8 could support SINV infection. We agree that this is a fascinating question to investigate but addressing it would probably require in depth analysis of tissue specific expression patterns of VLDLR, ApoER2, and MXRA8 in avian hosts along with affinity measurement of receptor fragments with the SINV spike protein that are beyond the scope of this current study.

Nonetheless, to mention the work by Zimmerman *et al.* and the notion that alphaviruses likely have complex patterns of interaction with different receptors in different hosts, we have added the following text to the manuscript (Lines 444–449):

“Interestingly, alphaviruses belonging to the western equine encephalitis complex, including SINV, were recently shown to recognize avian, but not mammalian, orthologs of MXRA8, the well-established CHIKV receptor¹¹, with a binding mode that is remarkably distinct from how CHIKV binds mammalian MXRA8⁴². These findings, along with our study, add to a growing understanding of the substantial structural plasticity through which alphavirus spike proteins can interact with cellular receptors on host cells^{18,19,36,42}.”

2. There are older data indicating that SFV receptor binding is not calcium dependent. Those studies did not use cells shown to be strongly dependent on expression of VLDLR, so HSPG binding may have played an important role. Can the authors comment on whether SFV or EEEV binding to VLDLR is inhibited by calcium chelation in any of their systems?

We thank the reviewer for bringing up this point, which was also brought up by reviewer #3 below. To address this point, we have carried out experiments with biolayer interferometry to test whether calcium chelation with EDTA inhibits EEEV and SFV binding to VLDLR. These experiments demonstrate that calcium chelation prevents a VLDLR ligand-binding domain Fc fusion protein (VLDLR_{LD}-Fc) from binding to EEEV and SFV VLPs (Supplementary Fig. 10). We believe this is because the calcium ion plays an integral role in stabilizing the fold of the LA repeats and in appropriately positioning the critical calcium coordinating acidic residues for productive interactions with basic residues on the alphavirus spike proteins.

A new section entitled “Calcium ions are necessary for VLDLR binding by EEEV and SFV” is included in the revised manuscript (lines 315–328).

As the reviewer notes, there is prior literature suggesting that infection of Vero cells by SFV does not depend on Ca^{2+} . In our search of the literature, we found two studies: Dubé *et al.* PLoS Pathogens 2014 and Dubé *et al.* Journal of Virology 2016. In both studies, experiments with SFV usually involved a pre-binding step, whereby virions were incubated with cells prior to modulation of Ca^{2+} levels in media, or addition of cell permeable or impermeable chelating agents. A key difference with these prior studies however is that they did not examine the role of Ca^{2+} in the initial receptor-attachment/binding event, which is the step our biolayer interferometry experiments with Fc fusion proteins and VLPs aim to mimic. The Ca^{2+} ions that are present in the VLDLR ligand-binding domain LA repeats are tightly bound and critical to protein folding in the endoplasmic reticulum, and they remain bound tightly at neutral pH. We thus suspect that very high concentrations of chelating agents would need to be used in experiments with cells to observe an effect.

3. I thought that the last paragraph of the discussion raised very interesting points. It would be helpful to the general reader to explain what is meant by the open vs. closed conformation of the LDLR family ligand binding domain.

Certain LDLR-related family proteins can adopt a closed conformation under acidic pH in which the ligand-binding domain folds back onto the EGF-like repeats or the β -propellers (PMIDs: 12459547, 36750096). In the closed conformation of such receptors, basic residues that are found in the EGF-like repeat/ β -propellers interact with the calcium ion-coordinating acidic residues on the LA repeats. It is thought that the propensity to adopt a closed conformation promotes cargo drop-off by replacing interactions LA repeats would otherwise be making with ligands. After reconsideration, we decided to remove our discussions about the closed conformations of LDLR-related receptors, because there are examples of alphavirus receptors like LDLRAD3 (the Venezuelan equine encephalitis virus receptor) that cannot adopt a closed conformation because they do not contain EGF-like repeats or β -propeller domains (see Figure 1a for a schematic representation of LDLRAD3).

4. Can the authors please comment on the technical differences that enabled them to obtain the structure of EEEV VLP with the VLRLR LA domain while the Cao paper reported that they could not? Maybe I missed this in the manuscript, but if not it would be helpful to point this out for the field.

We thank the referee for bringing up this important point. We removed this comparison because since the time we submitted our manuscript, additional groups including the authors from the Cao *et al.* Cell 2023 study we referenced (PMID: 37098345) have been able to also determine structures of VLDLR bound to EEEV a pre-print (<https://doi.org/10.11/2023.11.30.569340>). In their prior study, the authors mentioned that when they attempted cryo-EM analysis of EEEV in complex with ApoER2, they did not observe densities for the receptor. We believe that technical limitations including a potentially lower affinity of ApoER2 for EEEV may have explain these findings.

The new Cao *et al.* VLDLR EEEV pre-print in addition to a third study, first pre-printed and now published, Adams *et al.* Cell 2024 (PMID: 38176410) are discussed in lines 249–270 of our revised manuscript. This comparison describes how the structures are in overall agreement, although our structure provides additional information as our study is the only one to suggest a role for the presence of E3 in modulating receptor binding for VLDLR, which is a phenomenon that has also been described in structural analysis of CHIKV bound to its receptor, MXRA8.

Minor points:

1. I was puzzled by the inclusion of polybrene in the reporter virus entry studies. Please comment on this and whether it was required.

We thank the reviewer for raising this important point. While we had historically used polybrene in our experiments to increase adsorption on target cell membranes based on our experience with lentivirus transduction, we have found that the presence of polybrene does not affect the results of our experiments with RVPs in K562 cells. Therefore, for experiments we did during revisions, we did not include polybrene. We now specify in the methods section which experiments were performed with or without polybrene.

2. Please specify the cut-off for positive in Fig. 1b.

The cut-off for positive outcomes that are described in Fig. 1b were determined by whether a single LA repeat construct allowed entry into K562 cells that was higher and statistically significant as compared to RVP entry in K562 cells expressing the Δ LBD control as shown in Supplementary Data Figure 1d–g. We have added this information to the legend of Figure 1.

3. Please define cyan highlights in Fig. 2a.

Cyan background highlights residues completely conserved in all aligned LA repeats. We have added this information to the figure legend of Figure 2 and Supplementary Data Fig. 11c, d, and now cite the program used to generate the alignment (EScript 3) in all cases.

Reviewer #2 (Remarks to the Author):

The manuscript from Yang et al reports thorough, comparative structure-function studies of VLDLR recognition by EEEV and SFV. Cell-surface binding studies are used to identify the specific VLDLR LA repeats that EEEV and SFV bind. Cryo-EM structures are determined of EEEV and SFV VLPs in complex with soluble receptor constructs. Interestingly, EEEV and SFV have related but different modes of receptor recognition: VLDLR interacts with two basic residues on the EEEV E2 subunit whereas it interacts with two basic residues on the SFV E1 subunit. The general mode of binding is conserved among viral proteins and endogenous ligands that interact with LDLR-related proteins. The authors validate their structural findings with site-directed mutagenesis studies, which are well controlled. The results add to the growing body of knowledge concerning the interactions of viral and host proteins with LDLR-related proteins, with implications for viral evolution and adaptation.

Overall this is a very well written and presented manuscript with findings that will be of broad interest to virologists and structural biologists. The experiments are performed to a high standard, and the conclusions are justified by the data.

The authors should, however, show one or more supplementary figures/panels that display the EM map and model of each interface. The resolutions of the complexes are in the low-to-mid 3Å, and as the authors note they contain some heterogeneity due to averaging of different LA repeats. Thus, it is difficult to determine the confidence of side chains and hydrogen bonds/salt bridges that are shown in various figures (2b,e; 3a). There is some attempt to show this in Extended Data Fig 3 and 9, but the images are small and it

is difficult to tell whether sidechains are within the map. Some density for the EM maps is shown in Extended Data Fig 4, but these are not interface residues, so the value of showing this is rather limited.

We thank the reviewer for their overall positive assessment of our manuscript and for bringing up this point. We have included EM maps showing interface residues in Supplementary Figures 3 (bottom left) and 9 (bottom right). We had showed examples for EM density to highlight that various portions of the maps were of very high quality. Because of heterogeneity, the receptor LA repeat densities that we observed were less strong in quality, but our model building nonetheless relied on reliable density for positioning of backbone atoms and side chains that are bulky (e.g., the LA repeat tryptophan residue and the basic residues on the spike protein) into the EM density. We also placed side chains into density while considering expected geometrical and chemical properties. This approach allowed us to generate a model that we then experimentally validated using our mutagenesis studies in Figures 2 and 3.

Reviewer #3 (Remarks to the Author):

Comments for Authors

Semliki forest virus (SFV) and eastern equine encephalitis virus (EEEV) are two alphaviruses that can cause disease in humans. EEEV is of special clinical concern due to its high case fatality rate, the risk of neurological sequelae, and the lack of approved therapeutics or vaccines. In this manuscript, Yang and Li et al. report the structural basis for binding of SFV and EEEV to their cellular receptor VLDLR. Using single-particle cryo-electron microscopy the authors identified two separate binding modes for EEEV and SFV with the ligand-binding domain of VLDLR: EEEV engages VLDLR via the viral glycoprotein E2; SFV via the viral fusion protein E1. The structural observations are functionally validated in cellulo, using a structurally informed set of virus and VLDLR mutant constructs. Importantly, while cryo-EM structures of other alphavirus-receptor pairs are already available, they cannot be extrapolated to SFV and EEEV. This scientifically well-designed study therefore contributes a novel structural explanation of differential receptor-usage in the family of alphaviruses.

Major comments

1. The structural data shows that the calcium coordinating amino acids in the LA repeats are essential for interaction with both EEEV and SFV. SFV cell entry is not dependent on calcium and proceeds unperturbed even in the presence of a calcium chelator (e.g. Dube et al. 2016) – do the authors have any information on the effect of calcium chelators on VLDLR?

We thank the reviewer for bringing up this important point, which was also mentioned by reviewer #1 (see point 2) with the studies performed by Dubé *et al.* also discussed. To address this point, we have carried out experiments with biolayer interferometry and determined that calcium chelation prevents VLDLR_{LBD}-Fc from binding to immobilized EEEV and SFV VLPs. These data are included in a new Supplementary Figure 10. We used in these experiments as a control a VLDLR ligand antagonist (the receptor-associated protein, RAP) to show that adding RAP blocks VLDLR_{LBD}-Fc binding to SFV and EEEV VLPs. We also used Chikungunya virus (CHIKV) VLPs and an Fc-fusion version of the CHIKV receptor, MXRA8, to show that this interaction was not perturbed. A new section entitled “Calcium ions are necessary for VLDLR binding by EEEV and SFV” is included in the revised manuscript (lines 315–328).

Minor comments

1. Line 79-80: reference mix-up? Heidner et al. 1996 (Reference 2) is cited for equine infection with encephalitic alphaviruses in the Americas; data is however found in Corrin et al. 2021 (Reference 4).

We have corrected the references by removing Heidner *et al.* 1996. Corrin *et al.* 2021 is now reference 2.

2. Line 93-94: wording. “Importantly, while LDLRAD3 is a receptor for VEEV, it is not a receptor for any other [tested] alphavirus.”

We have corrected this statement (please see lines 92–93 in the revised manuscript).

Reviewer #4 (Remarks to the Author):

Previously, L. Clark and colleagues described very-low density lipoprotein receptor (VLDLR) as a receptor for certain alphaviruses such as eastern equine encephalitis virus (EEEV) and Semliki forest virus (SFV). It was shown that the VLDLR ligand-binding domain (LBD), which contains LDLR class A (LA) repeats, is needed to support alphavirus infection and that the viral envelope glycoproteins (E proteins) engage with the receptor. In this manuscript, Yang et al. determine the structural basis for binding of EEEV and SFV to VLDLR by cryo-electron microscopy (cryo-EM). While VLDLR contact sites between EEEV and SFV are divergent, the authors show that basic residues in the viral E proteins are required for interaction with the receptor. Using mutated E proteins and VLDLR LA repeat constructs, importance of contact sites in VLDLR and E proteins determined by cryo-EM was further validated by cell surface staining experiments and through infectivity assays.

Finally, the authors compared the identified binding mode to the interactions of LA repeats with other viruses and ligands.

In sum, the study gives important insights into the interaction of alphaviruses with lipoprotein receptors. The study is novel and original. References are appropriately cited. Overall, the manuscript is well written. Experiments are well explained and the technology used is highly innovative and suitable for the questions addressed. However, certain questions remain that need to be discussed before publishing the manuscript. Additionally, some changes have to be implemented into the manuscript. These are listed below.

Major remarks:

1. Statistics: In Fig. 2c, d, f, g and Extended Data Fig. 1d-g it is stated that two to three biological experiments were performed. However, it seems that values of technical replicates were used to perform statistical analyses (e.g. indicated by stating that $n=9$ in Extended Data Fig. 1 when only three experiments with three technical replicates were performed). Please re analyse the according data set and state $n=x$ correctly.

We have re-analyzed all data using means of technical replicates from each biological experiment instead of using the technical replicates themselves. Fig. 2c, d, f, g and Supplementary Figures 1d–g have been updated to show means of technical replicates. We also analyzed new data obtained during revisions in Figures 3d, e, and g in a similar manner. Figure legends have been updated to reflect this change. Re-analysis did not affect the significance of the observations nor their conclusions.

2. Extended Data Fig. 1d & Extended Data Fig. 2c & d: K562 cells expressing VLDLR LA1 seemed to be highly permissive to EEEV RVP, whereas biolayer interferometry experiments indicated low affinity of VLDLRLA1-Fc binding to EEEV VLPs. Please discuss possible explanations for these observations.

We believe these findings are related to avidity. In the biolayer interferometry experiments, we coated biosensors with EEEV VLPs and allowed VLDLR_{LA1}-Fc in solution to associate with the VLPs. In this experimental set-up, VLDLR_{LA1}-Fc is free to associate and dissociate in a manner that is independent of the binding events on other spike protein subunits of the same VLP. This is reflected by a very high off rate. In K562 infectivity assays, however, EEEV virions can simultaneously engage multiple copies of the single VLDLR LA1 receptor construct on the cell surface, which effectively reduces the off rate and should promote a highly avid binding mode. In this case, though a single LA1 repeat may have a low affinity for EEEV, binding multiple copies of LA1 on K562 cells facilitates avid binding.

Minor remarks:

1. A VLDLRLBD-Fc construct is used throughout the study. It would be nice to illustrate the structure of the construct as done in Fig. 1a for other constructs.

Schematic illustrations for VLDLR_{LBD}-Fc, VLDLR_{LA1}-Fc, and MXRA8_{ect}-Fc have been added to Supplementary Data Figure 2a–c.

2. Color code: Please make sure to explain the color code used in figures; especially colors used in alignments were not explained properly.

We have modified relevant figure legends to address color coding in figures and alignments. Fig 2a and Supplementary Data Fig. 11c, d: Cyan background highlights residues completely conserved in all LA repeats aligned. Boxed residues highlight positions where a single majority residue or multiple chemically similar residues could be identified. Such residues are highlighted in magenta. Supplementary Data Figure 8: Red background highlights residues completely conserved in all sequences aligned. Boxed residues highlight positions where a single majority residue or multiple chemically similar residues could be identified. These residues are highlighted in red. We also now cite the software we used to generate the sequence alignment (ESPrnt 3) in which these representations are standardly used.

3. Extended Data Fig. 1: The expression of LA2Flag in K562 seemed to be lower than the expression of other constructs (see Extended Data Fig. 1a). Please comment if this could have an influence on infection with tested RVPs (e.g. VEEV RVPs did not seem to be able to engage with LA2 under tested conditions).

We agree with the reviewer that the single LA2 Flag construct expressed to lower levels than the other single LA repeat constructs; however, most LA2 Flag cells nonetheless had stronger staining signals than the isotype control antibody-stained cells, indicating that most cells express

the construct. Furthermore, this level of LA2 Flag construct expression was sufficient for this construct to support robust infection by SFV and EEEV RVPs (see Supplementary Figure 1d, e). We also observed a statistically significant phenotype for SINV, which probably binds LA repeats less efficiently (Supplementary Figure 1f). Observing a range of phenotypes with SFV, EEEV, and SINV RVPs but no positive cells with VEEV RVPs, makes us believe that VEEV RVPs are unlikely to bind LA2.

We have modified the text as follows based on the reviewer comment (see underlined portions):

“We overexpressed these truncated VLDLR variants or full length VLDLR in K562 cells, a human lymphoblast cell line that does not express VLDLR or ApoER2¹, and confirmed cell surface expression of constructs using immunostaining, although the single repeat LA2 construct had lower expression (Supplementary Fig. 1a, b).”

“We used EEEV, SFV, and SINV RVPs that express green fluorescent protein (GFP) to infect K562 cells transduced with single LA repeat constructs and found that each RVP differed in the repeats it could recognize to infect cells under tested conditions (Fig. 1b, Supplementary Fig. 1c–f).”

4. The authors show that only small conformational changes occur in the EEEV glycoprotein upon VLDLR binding. Additionally, they mention that “some LDLR-family receptors ... favor a closed conformation at low pH“. Would it be possible to modulate changes in viral E proteins and VLDLR at low pH as compared to neutral pH (e.g. by computational modulation)?

We agree with the reviewer that additional clarification would be needed here. The potential shift from an open to a closed conformation by VLDLR under acidic pH has not been observed, but such a shift has been structurally observed for the closely related proteins LDLR and LRP2 (PMIDs: 12459547 and 36750096). In the closed conformations of these receptors, the ligand-binding domain folds back onto the EGF-like domains or the β -propeller domains in a manner that occludes the LA-repeat ligand-binding surface to promote cargo release. We suspect that at low pH, full-length VLDLR will adopt a closed conformation, promoting its dissociation from alphavirus spike protein. Furthermore, the low pH of the endosome decreases the affinity of the LA repeats for Ca^{2+} by several orders of magnitude, and the endosome also has low-free calcium levels, which probably also contribute to cargo release (PMID: 19583244). We would speculate that closure of VLDLR upon exposure to low pH could promote dissociation of the LA repeat from the alphavirus spike protein binding sites through this closure event (which, as noted above, would occlude LA-repeat ligand binding surfaces), and also perhaps through acid dependent effects on Ca^{2+} binding.

The conformational changes that alphavirus spike proteins undergo as the spike proteins are exposed to low pH involves an “opening” of the trimeric spike in which the E2 glycoprotein partially dissociates from E1, exposing the fusion loops at the tip of E1, followed by a transition to a closed, E1 trimeric state that coincides with membrane merger. Two seminal studies described some of these transitions (PMIDs: 21124457, 21124458). Whether this conformational change in the spike protein would occur with the same kinetics as closure of VLDLR is unknown but could be an interesting process to model computationally; however, we do not have access to the tools yet to do so (e.g., no structures of the fully open and closed VLDLR).

These are important points, but due to their speculative nature and based on a question that arose from another reviewer (see Reviewer #1 point 3), we have decided to remove from the manuscript the point about closure of VLDLR being related to virus release.

5. Extended Data Fig. 5: Some residues that are mentioned in the results part are not shown in the figure (e.g. H155). Please add them.

We have added and labeled H155 in Extended Data 5a, c (Now renamed Supplementary Figure 5a, c).

6. Page 9: The authors state that “...EEEV can engage LA repeats that contain either a tryptophan or a phenylalanine as the key aromatic residue. However, the LA2 W89F mutation abolished SFV RVP entry into K562 cells (Fig. 2f). These findings suggest that the polar contact the SFV E1 D327 side chain makes with the indole nitrogen of the LA repeat tryptophan (Fig. 2e) is a critical interaction and highlight a strict requirement of a tryptophan as the LA repeat aromatic residue for engagement of the SFV spike protein.”. Wouldn't it be theoretically possible that another aromatic residue than phenylalanine could also support SFV infection? If this is the case, please re-phrase the sentence.

We thank the reviewer for this comment, as it would indeed be possible for another aromatic residue to potentially make this contact (e.g., tyrosine or histidine). Within LA repeats, tyrosines as the aromatic residue are exceedingly rare, and to the best of our knowledge, histidines are not usually found in this position. However, as shown in the sequence alignment in Figure 2a, other polar residues can at times occupy the position that is usually an aromatic residue in LA repeats (e.g., LA7 and LA8 are R and K, respectively). To avoid confusion, we have removed the portion of the sentence that was problematic and now write (lines 244–246):

“These findings suggest that the polar contact the SFV E1 D327 side chain makes with the indole nitrogen of the LA repeat tryptophan (Fig. 2e) is a critical interaction.”

7. The authors should consider adding parts of their supplementary figures to the main text (e.g. Extended Data Fig. 1d-g and Extended Data Fig. 2 c).

We considered this suggestion, but to avoid overcrowding the main figures while incorporating other revisions, and to maintain an emphasis on the structural data, we decided to leave Extended Data Fig. 1d–g and Extended Data Fig. 2c (now Supplementary Figure 2e, f) in the supplemental materials.

8. Performing infection assays with EEEV and SFV harboring E proteins with mutated VLDLR contact sites could further support the findings of this manuscript.

We thank the reviewer for making this suggestion. In the revised manuscript, we now include infectivity assays in which we mutated the EEEV and SFV E proteins to confirm the VLDLR contact sites (e.g., EEEV E2 K156A and SFV E1 K345A). These new data are now provided in Figure 3 (for the EEEV data) and Supplementary Fig. 7f for the SFV data, with infectivity assays performed on K562 cells expressing VLDLR and also a shorter isoform of ApoER2 (iso2), which we previously showed could serve as a receptor for EEEV and SFV (Clark *et al.* 2022). These new data further support the cryo-EM structures by showing that EEEV E2 K156 and SFV E1 K345 are critical determinants of binding to not only VLDLR but also to the structurally homologous receptor, ApoER2. Please note that we found that EEEV mutant RVPs and SFV

mutant RVPs retained infectivity and could be tittered on Vero E6 cells and C6/36 (mosquito) cells, respectively, as shown in titers provided in Supplementary Fig. 7f.

Since our manuscript was submitted and our pre-print was posted, two other studies (one now published and another posted as a pre-print) have identified two additional VLDLR binding sites on the EEEV spike protein (Adams et al. *Cell* 2024, PMID: 38176410, and Cao et al. (<https://doi.org/10.11/2023.11.30.569340>). While the LA repeat binding site we described in our study that is within the cleft formed by adjacent E2–E1 protomers is “site 1,” the other studies reported additional sites in E2 domains A (“site 2”) and E2 domain B (“site 3”). Site 3 is very rare and has only been observed in two EEEV strains. Comparison with one of the structures is now provided in Figures 3a and 3b. We also discuss in the revised manuscript why we think we observed a single LA repeat binding mode while the other studies observed three LA repeats, in a section entitled “E3 hinders multiple LA repeat binding modes.”

We also used EEEV RVPs with both site 1 and site 2 mutations (E2 K231/K232) in infectivity assays on K562 cells expressing VLDLR and ApoER2, and for these experiments, we chose RVPs for an EEEV strain that has two LA repeat binding sites (FL91-469; this was the strain we used for all infectivity assays in our original submission) and we also generated RVPs for a strain that has three sites (PE6, which is the strain we used for structural analysis with VLPs). We found, using those RVPs, that in a strain that only has two sites (FL91-469), site 1 is critical for binding to VLDLR and ApoER2 (Fig. 3d). We also found that for a strain that has three sites (PE6), site 1 is not required for binding to VLDLR but is still important for binding to ApoER2 (Fig. 3e). We note that while Adams *et al.* had also performed mutagenesis assays and found that site 1 can be dispensable for strains that have three binding sites, no other study had examined ApoER2 binding. Our structural and functional analysis thus suggest that for a strain that, like most circulating strains, only has two sites, site 1 is nonetheless a critical site for receptor binding, and that for a receptor that has a shorter ligand-binding domain (e.g., ApoER2 iso2), EEEV primarily relies on site 1 for binding.

Lastly, to further increase the significance of our findings as compared to the other published/pre—printed studies, we have also included infectivity assays with murine primary neurons to show that both sites 1 and 2 are important for infection of this physiologically relevant cell type (Fig. 3f, g).

The new findings are described in sections entitled “Spike protein determinants of VLDLR and ApoER2 binding” lines 272–302 and “LA repeat binding sites are determinants of neuronal entry” lines 304–313.

9. Extended Data Table 1: The EEEV VLP:VLDLRLBD-Fc defocus range given in the table differs from the one given in the methods part. Please correct.

We thank the reviewer for bringing up this mistake and we have corrected the table to include the correct defocus range we used during collection of the EEEV VLP:VLDLRLBD-Fc dataset.

10. Please correct grammatical errors and spelling mistakes; a few times the font size/spacing changes, in Extended Data Figure 1c the labeling is shifted.

We have corrected grammatical errors and spelling mistakes throughout the text and fixed the font size/spacing issues. We have also corrected the shifted label of Extended Data Figure 1c, which is now Figure S1c.

REVIEWERS' COMMENTS

Reviewer #1 (Remarks to the Author):

This revised manuscript by Yang et al. has responded well to the reviewer comments. In particular, the addition of experiments to address calcium requirements for VLDLR binding and the addition of mutagenesis studies to test critical residues adds important information to the paper. The paper was already strong and the revisions have improved it. I have no further comments.

Reviewer #2 (Remarks to the Author):

The authors have adequately addressed my comments.

Reviewer #3 (Remarks to the Author):

Remarks to the authors

The reviewer comments for the original Yang and Li et al. submission “Structural basis for VLDLR recognition by eastern equine encephalitis virus” were generally enthusiastic and highly favourable. However, several major and minor concerns were raised by the 4 reviewers, including the suggestion to discuss (1) recently published / pre-printed alphavirus-receptor structural studies (Adams et al. Cell 2024; Cao et al. preprint; Zimmerman et al. 2023), (2) the role of Ca²⁺ in receptor-virus engagement given that SFV entry was previously found to be Ca²⁺ independent, (3) the apparent discrepancy between in cellulo observations and BLI data (extended Figures 1D, 2C and 2D), and (4) to correct improper statistical analysis where technical repeats rather than means of each biological experiment were compared.

In this current revised manuscript, Yang and Li et al. adequately addressed all major and minor reviewer comments. Of note, the authors included new infectivity assays with mutated EEEV and SFV E proteins (Figure 3 and Extended Figure 7F). This new data not only strongly supports the relevance of the respective residues in VLDLR / ApoER2 receptor

engagement but also provides a model to explain different/additional binding sites of VLDLR on the EEEV spike complex that were recently reported by two other groups (cf. major concern (1) above). As no additional concerns were raised, I therefore recommend this revised manuscript for publication.

Reviewer #4 (Remarks to the Author):

All issues raised by the reviewers have been addressed to full satisfaction. The study, despite additional structural studies being published in the meantime, is a seminal one, which provides important knowledge to the scientific community. The findings are of broad interest, and the study is of high quality.